# A translational MRI approach to validate acute axonal damage detection as an early event in multiple sclerosis

**Antonio Cerdán Cerdá**[1†], **Nicola Toschi**[2,3†], **Constantina A Treaba**[2], **Valeria Barletta**[2], **Elena Herranz**[2], **Ambica Mehndiratta**[2], **Jose A Gomez-Sanchez**[1,4,5], **Caterina Mainero**[2‡], **Silvia De Santis**[1*‡]

[1]Instituto de Neurociencias de Alicante, CSIC-UMH, San Juan de Alicante, Spain; [2]Athinoula A. Martinos Center for Biomedical Imaging, Department of Radiology, Massachusetts General Hospital, Harvard Medical School, Boston, United States; [3]Department of Biomedicine and Prevention, University of Rome Tor Vergata, Rome, Italy; [4]Instituto de Investigación Sanitaria y Biomédica de Alicante (ISABIAL), Alicante, Spain; [5]Millennium Nucleus for the Study of Pain (MiNuSPain), Santiago, Chile

*For correspondence:
dsilvia@umh.es

†These authors contributed equally to this work
‡These authors also contributed equally to this work

Competing interest: The authors declare that no competing interests exist.

**Abstract** Axonal degeneration is a central pathological feature of multiple sclerosis and is closely associated with irreversible clinical disability. Current noninvasive methods to detect axonal damage in vivo are limited in their specificity and clinical applicability, and by the lack of proper validation. We aimed to validate an MRI framework based on multicompartment modeling of the diffusion signal (AxCaliber) in rats in the presence of axonal pathology, achieved through injection of a neurotoxin damaging the neuronal terminal of axons. We then applied the same MRI protocol to map axonal integrity in the brain of multiple sclerosis relapsing-remitting patients and age-matched healthy controls. AxCaliber is sensitive to acute axonal damage in rats, as demonstrated by a significant increase in the mean axonal caliber along the targeted tract, which correlated with neurofilament staining. Electron microscopy confirmed that increased mean axonal diameter is associated with acute axonal pathology. In humans with multiple sclerosis, we uncovered a diffuse increase in mean axonal caliber in most areas of the normal-appearing white matter, preferentially affecting patients with short disease duration. Our results demonstrate that MRI-based axonal diameter mapping is a sensitive and specific imaging biomarker that links noninvasive imaging contrasts with the underlying biological substrate, uncovering generalized axonal damage in multiple sclerosis as an early event.

## Editor's evaluation

This is a valuable study that aims to validate and translate an established non-invasive proxy measure of axonal diameter that is derived from magnetic resonance imaging. The results are solid, demonstrating alterations in the proxy measure in rodent models of axonal damage and patients with multiple sclerosis. The Discussion acknowledges weaknesses relating to the details of modelling and signal-to-noise ratio of the measurements. This work will be of interest to researchers studying the microstructural changes in neurodegeneration.

## Introduction

Axonal damage is the main pathological substrate of irreversible neurological disability in multiple sclerosis (MS). In MS, axonal damage can either be direct or secondary to demyelination, glial

activation, or exposure to excitatory amino acids and cytokines (*Haines et al., 2011*). As the transition to progressive MS occurs when an axonal loss threshold is reached, and the brain compensatory capacity is surpassed (*Criste et al., 2014*), the development of novel in vivo, noninvasive strategies for characterizing axonal microstructure becomes essential for early disease detection.

Magnetic resonance imaging (MRI), particularly diffusion-based approaches, has been applied in MS to investigate the white matter (WM) damage (*Inglese and Bester, 2010*). However, conventional diffusion MRI is unspecific to different tissue compartments, such as axons or myelin (*De Santis et al., 2014*), hampering the ability to distinguish axonal damage from other microstructural changes. In addition, due to the limitations of quantitatively comparing in vivo to ex vivo data, validation of imaging findings is rarely performed (*Horowitz et al., 2015*). While animal models recapitulating specific aspects of MS pathophysiology are available (*Torkildsen et al., 2008*, *Constantinescu et al., 2011*), none focuses on axonal damage but rather on the demyelinating aspect of the disease.

AxCaliber is an advanced imaging framework able to estimate an MRI axonal diameter proxy (*Assaf et al., 2008*), which has become applicable in humans thanks to recent hardware advances (*Jones et al., 2018*). AxCaliber has revealed a higher axonal diameter proxy in the normal-appearing WM (NAWM) of the corpus callosum of MS patients compared to healthy controls, which was interpreted as a sign of axonal damage (*Huang et al., 2016*). However, in its original formulation, AxCaliber is only applicable to voxels characterized by a single fiber orientation (such as the corpus callosum, as in *Huang et al., 2016*), while at least 70% of brain voxels contain two or more dominant fiber orientations (*Jeurissen et al., 2013*). Recently, we used a modified AxCaliber model to map whole-brain MRI axonal diameter proxy (*De Santis et al., 2019b*), hence providing a means to characterize the axonal damage in MS in a whole-brain fashion.

From a pathophysiological point of view, measuring a higher estimated MRI axonal diameter proxy in MS does not, per se, represent conclusive evidence of axonal damage, especially in the light of the known bias toward larger axons of MRI-based axonal diameter quantification (*Horowitz et al., 2015*). Indeed, MRI sensitivity to axonal pathology is yet to be fully demonstrated to establish its clinical utility. Recently, an approach to selectively manipulate specific microstructural aspects of the parenchyma through targeted, unilateral injection of neurotoxic agents has been proposed (*Garcia-Hernandez et al., 2022*).

In this study, we aimed to (i) validate AxCaliber-based axonal mapping in a preclinical model of fimbrial damage induced by stereotaxic injection of ibotenic acid into the hippocampus and (ii) translate the model to investigate changes in axonal microstructure in the brain white matter of patients with MS, particularly in normal-appearing tissue where pathological microstructural abnormalities have been previously reported (*Luchicchi et al., 2021*). Specifically, we (iia) compared whole-brain, voxelwise MRI axonal diameter proxy in MS patients relative to healthy controls and (iib) tested the association between clinical features and the MRI axonal diameter proxy in MS patients. We uncovered an abnormal increase in the MRI axonal diameter proxy in NAWM in MS, outside the inflammatory lesions, which was greater in patients with short disease duration, implicating axonal pathology as a primary event in MS pathogenesis.

## Results

### A rat model of acute axonal damage

Fourteen days after injection of the neurotoxin ibotenic acid into the hippocampus, paired t-test revealed a significant increase in the mean MRI axonal diameter proxy (p=0.021) in the fimbria belonging to the injected hemisphere compared to the control (*Figure 1e*), confirming that the damage affected a large portion of the tract. The other parameters extracted from the MRI analysis are not significantly different between hemispheres, although there is a tendency of reduced slope of the extra-axonal radial diffusivity decay for increasing diffusion time in the injected hemisphere (*Figure 1—figure supplement 1*). The fimbrias were reconstructed through Diffusion Tensor Imaging (DTI)-based tractography. Through tract-based analysis, we revealed a significant effect of the injection ($F_{1,9}$=20.3, p=0.001), of the position along the tract ($F_{48,432}$=83.9, p<0.001) and of their interaction ($F_{48,432}$=5.7, p=0.003); post-hoc comparisons between injection type, performed for each position and corrected for multiple comparisons, revealed significant differences in the mean MRI axonal diameter proxy between ibotenic acid- and saline-injected tracts in most parts of the tract. Significant

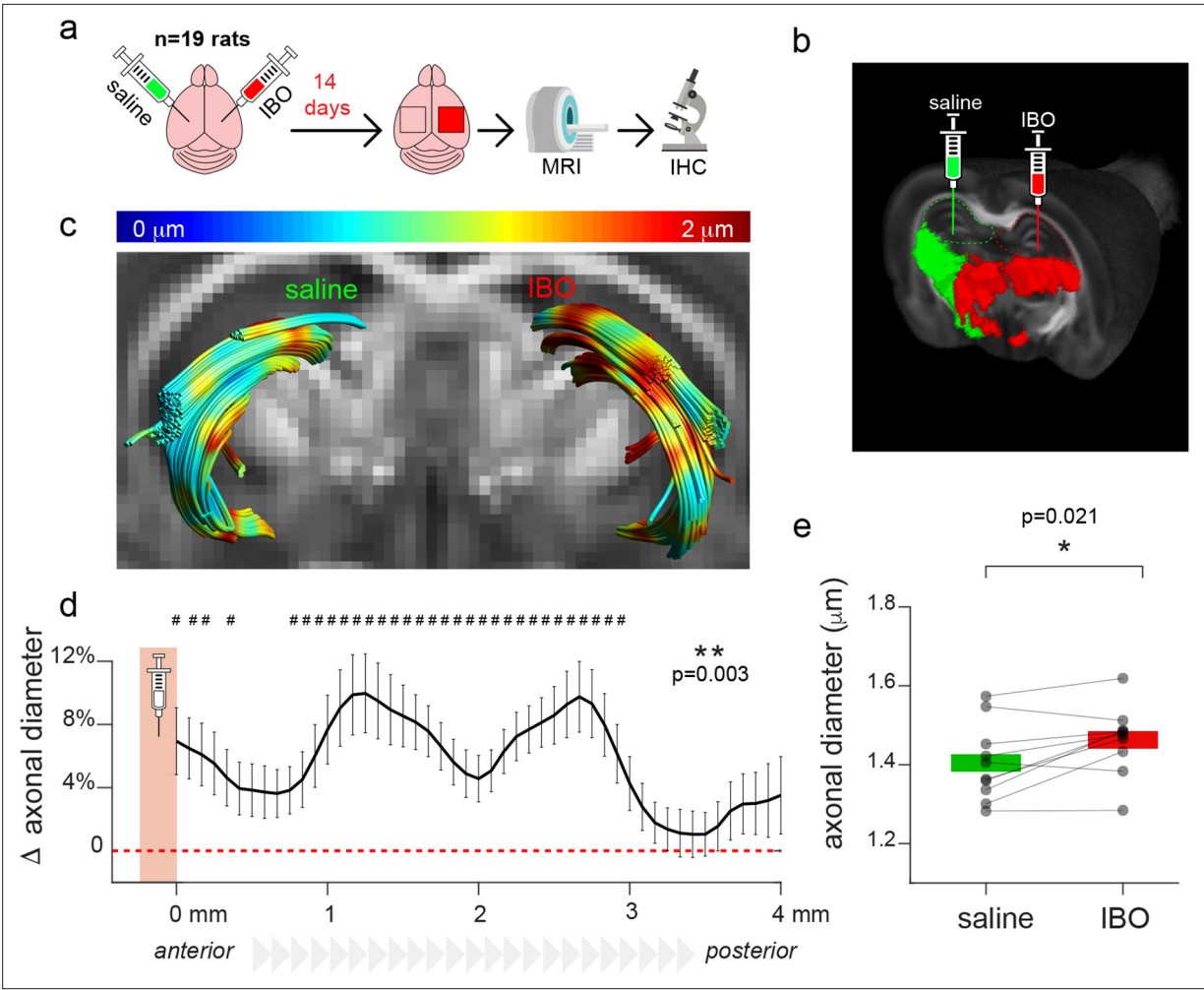

**Figure 1.** Experimental model of axonal damage. (**a**) Experimental scheme of stereotaxic injections of ibotenic acid (IBO) in the left hippocampus of n=19 rats. The right hippocampus was injected with saline solution and used as a control. (**b**) Visualization of the injection setup. (**c**) Example of the tractography of the fimbriae from one representative animal, superimposed on the fractional anisotropy map. The MRI axonal diameter proxy is projected on the tract through color coding. (**d**) Mean difference and standard deviation between groups of MRI axonal diameter proxy measured across all the streamlines constituting the fimbria in the antero-posterior axis, starting from the injection point (n=10). The injection site is shown in red. Asterisks represent significant group effect in the ANOVA, while hashtags represent significant post-hoc differences between groups in each location, corrected for multiple comparisons. (**e**) Mean MRI axonal diameter proxy calculated in the ibotenic vs saline-injected fimbria reconstructed using tractography. Asterisks represent significant differences (n=10, paired t test across hemispheres, p=0.021).

The online version of this article includes the following figure supplement(s) for figure 1:

**Figure supplement 1.** Other MRI parameters in control vs injected fimbriae.

**Figure supplement 2.** Axonal diameter estimation using the low b-value MRI protocol.

**Figure supplement 3.** Comparison between linear and log(t)/t functional forms.

differences were mostly localized posterior to the injection site (*Figure 1d*). In the subset of animals undergoing a protocol with lower b-values and a diffusion model not including the Δ dependency (AxCaliber), we obtained similar results (*Figure 1—figure supplement 2*).

When comparing immunofluorescence staining in ibotenic acid- versus saline-injected hemispheres, we confirmed both neuronal loss in the hippocampus (p=0.026, *Figure 2a–b*) and axonal damage in the fimbria (p=0.047, *Figure 2c–d*), corresponding to a lower staining intensity of neuronal nuclear protein (NeuN) and higher intensity of neurofilament staining in the hemisphere injected with ibotenic acid. No differences were found in the fimbria myelin content using Myelin Basic Protein (MBP) staining (*Figure 2—figure supplement 1*), suggesting that at the studied time point, axonal structure was significantly altered, but the total myelin content was still preserved. Neurofilament fluorescence intensity

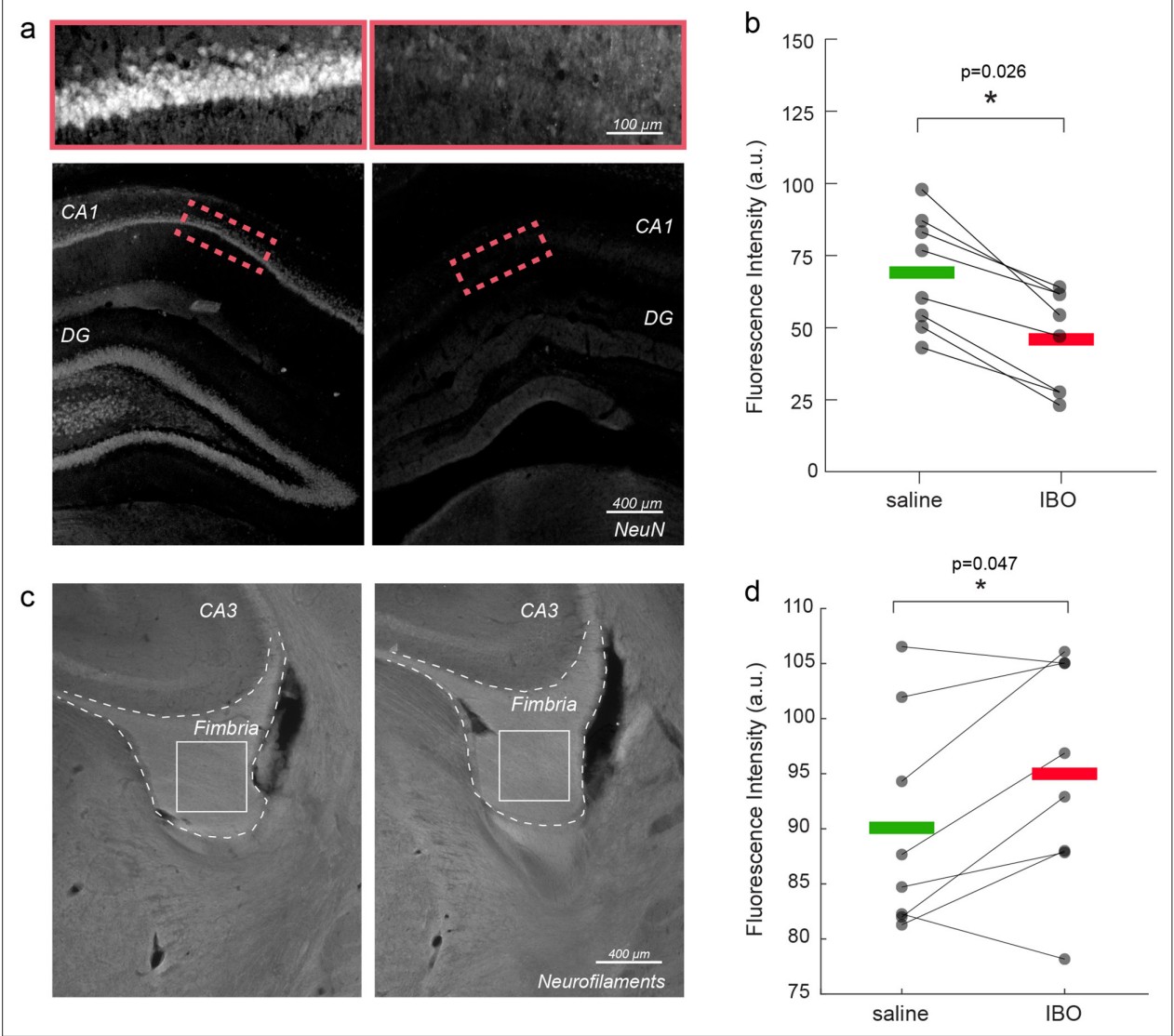

**Figure 2.** Immunofluorescence validation of axonal damage. (**a**) NeuN staining in control vs. injected hippocampi. (**b**) Mean NeuN intensity in control vs. injected hippocampi. Asterisks represent significant differences across hemispheres (n=8, paired t test, p=0.026). (**c**) Neurofilament staining in control vs. injected fimbria. (**d**) Mean neurofilament intensity in control vs. injected hippocampi. Asterisks represent significant differences in means across hemispheres (n=8, paired t test, p=0.047).

The online version of this article includes the following figure supplement(s) for figure 2:

**Figure supplement 1.** Myelin Basic Protein staining in injected versus control fimbria.

**Figure supplement 2.** Correlation between MRI and histology.

was significantly correlated with the axonal diameter proxy measured with MRI (*r*=0.54, p=0.029) in both the fimbria tract of injected and control hemispheres (*Figure 2—figure supplement 2*).

Scanning transmission electron microscopy (STEM) revealed increased axonal diameter in the hemisphere injected with ibotenic acid, with no significant reduction in axonal count, indicating limited axonal loss, as reported in *Figure 3*. In *Figure 3—figure supplement 1*, the total brain shrinkage from in vivo to after perfusion was quantified in three animals as 28%. Since the post-fixation with 1% osmium tetroxide gives at least a 15% additional shrinkage (*Kinney et al., 2013*), the total shrinkage caused by the STEM preparation can be quantified as 39%.

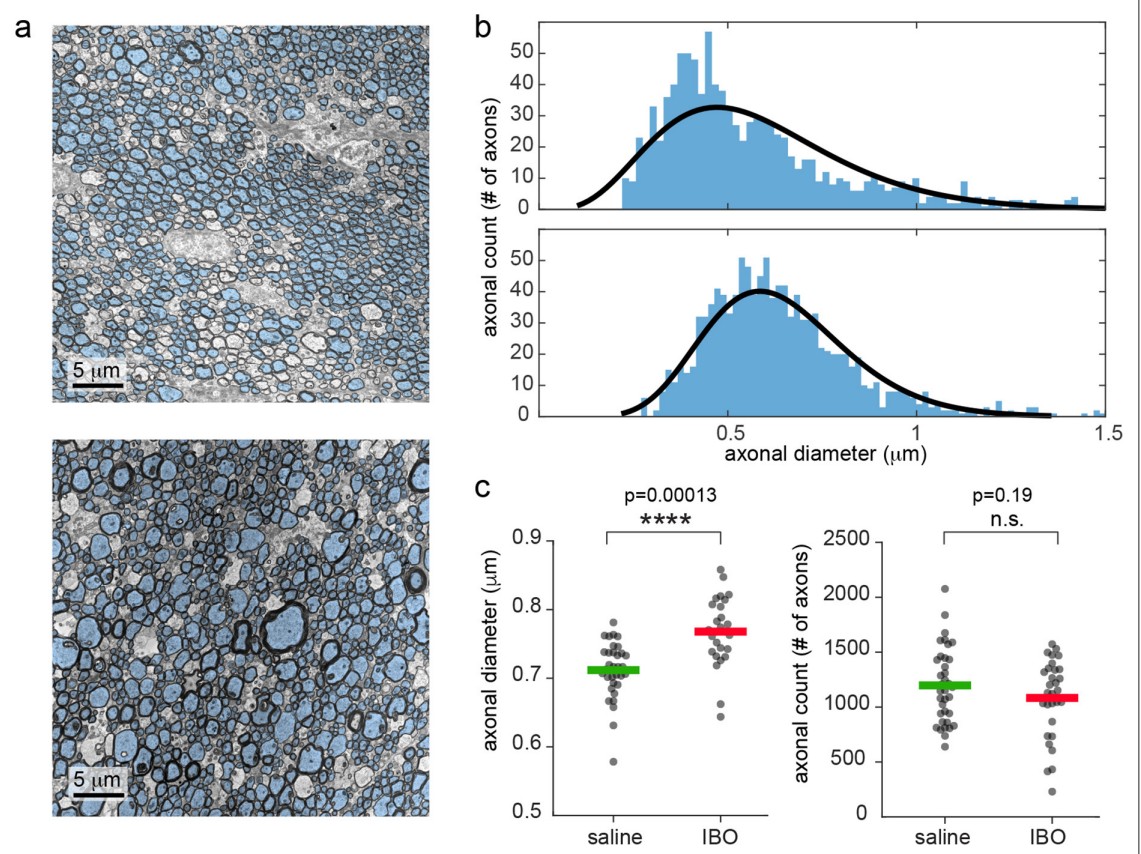

**Figure 3.** Electron microscopy shows increased mean axonal diameter in ibotenic-injected hemisphere compared to saline. (**a**) Representative STEM photos for saline and ibotenic acid fimbriae. Segmented axons are overlaid in light blue. (**b**) Histogram of the axonal count in one representative animal: upper line, saline injected, lower line, ibotenic. Black lines represent the gamma function better fitting the histogram. (**c**) Mean axonal diameter (left) and count (right) in each photo and group. Asterisks represent significant unpaired t test differences between groups for axonal diameter (n=6, p=0.00013).

The online version of this article includes the following figure supplement(s) for figure 3:

**Figure supplement 1.** Brain shrinkage during histology.

## Axonal damage in normal-appearing white matter of multiple sclerosis patients

After preclinical validation in rats, we applied the clinical version of the AxCaliber MRI protocol to a cohort of 11 MS patients and 10 age-matched healthy controls. When comparing the MRI axonal diameter proxy in the NAWM of MS patients and controls, we found higher values in the MS group (p<0.05, corrected; *Figure 4*). The differences were mostly symmetrical across hemispheres and involved all major WM tracts, notably: the corpus callosum, the corticospinal tract, the internal capsule, the corona radiata, the thalamic radiation, the inferior longitudinal fasciculus, the cingulum, the fornix, the superior longitudinal fasciculus, the inferior fronto-occipital fasciculus, the uncinate fasciculus, and the tapetum.

## Axonal diameter is preferentially increased in patients with early disease

Next, we tested for associations between the measured MRI axonal diameter proxy and the disease duration. The rationale is that axonal swelling could be an early event in the disease, as suggested by postmortem evidence (*Luchicchi et al., 2021*).

Tract-based spatial statistics unveiled a trend of negative correlation between the magnitude of the MRI axonal diameter proxy in patients and the disease duration, as shown in *Figure 5a*. When comparing average values of MRI axonal diameter proxy in the whole white matter (excluding lesions in MS) between groups (controls, short and long disease duration), we report a significant group effect

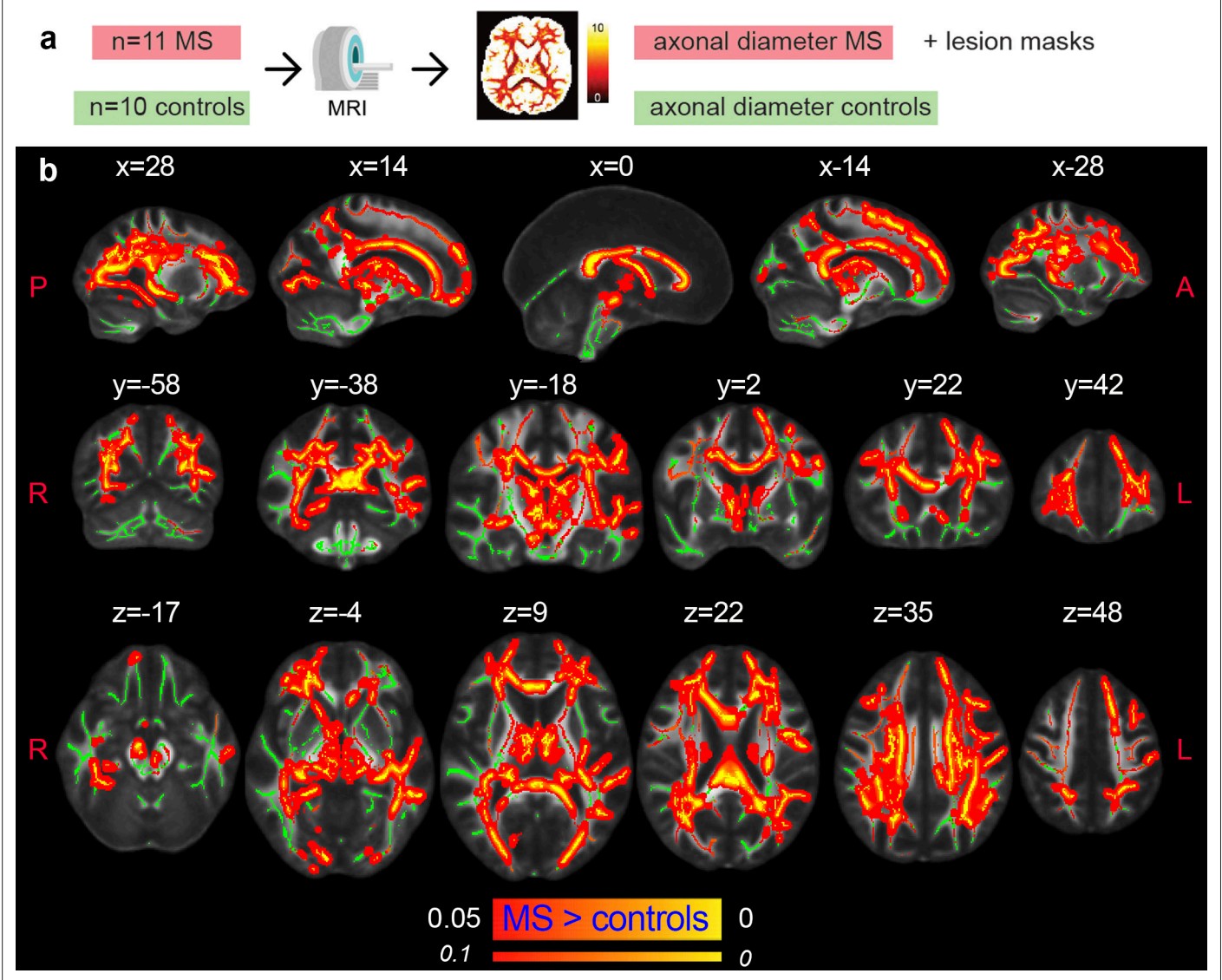

**Figure 4.** Axonal damage in MS normal-appearing white matter. (**a**) Experimental scheme. (**b**) Tract-based spatial statistics showing voxels in which the mean MRI axonal diameter proxy is significantly increased in multiple sclerosis versus healthy conditions (n=21, p<0.05, corrected). The opposite contrast was not statistically significant. Green: skeletonized white matter. Inflated red-yellow (through the pipeline tbss_fill): significant p value. Red-yellow: p-value <0.1.

The online version of this article includes the following figure supplement(s) for figure 4:

**Figure supplement 1.** Slope of extra-axonal radial diffusivity and restricted signal fraction in patients vs. controls.

**Figure supplement 2.** Rician simulations showing accuracy of MRI axonal diameters proxy.

in the ANOVA ($F_{2,18}$=9.2, p=0.002). MS patients for whom the disease onset was less than 5 years prior to the MRI scan had increased axonal diameter compared to controls (p=0.001, corrected for multiple comparisons), while this increase was not significant in MS patients with a longer disease course (p=0.18). We did not find significant associations between axonal diameter and other tested clinical variables of neurological disability and information processing speed (Expanded Disability Status Scale [EDDS] and Symbol Digit Modalities Test [SDMT]).

Finally, we tested both group differences and associations with disease duration for the rest of the parameters extracted in the MRI analysis. While both the slope of the extra-axonal radial diffusivity decay for increasing diffusion time and the restricted signal fraction are significantly reduced in a

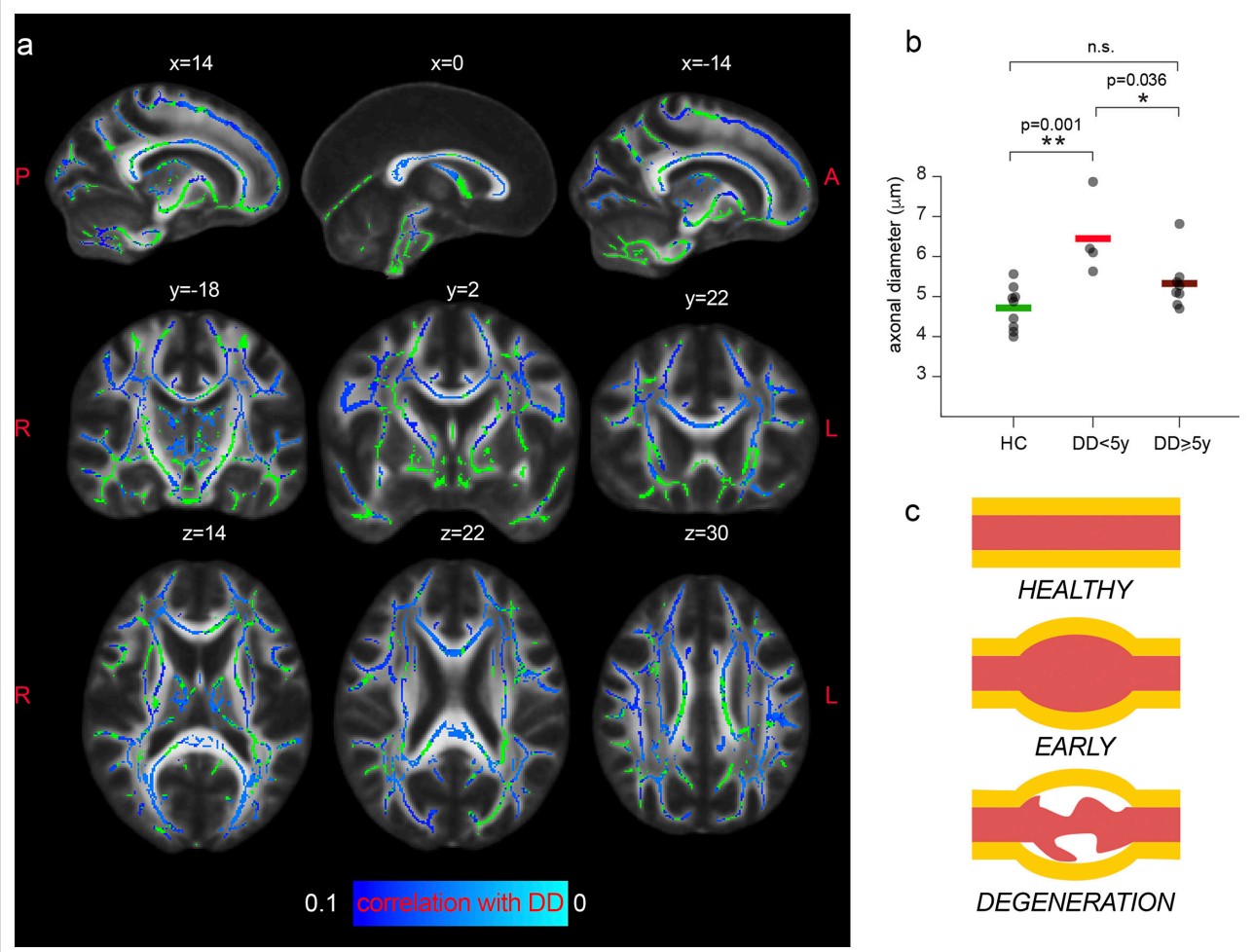

**Figure 5.** Axonal diameter is preferentially increased in patients with early MS. (**a**) Tract-based spatial statistics showing voxels in which a trend of negative association between the MRI axonal diameter proxy and the disease duration (DD) in patients is present (n=11, p<0.1; lowest p-value = 0.051 corrected). Green: skeletonized white matter. Bluel-light blue: p value. (**b**) Mean axonal diameter in the whole with matter of healthy controls (n=10, green), MS patients early in the disease course (n=4, <5 years, in red) and MS patients with a longer disease trajectory (n=7, ≥ 5 years, dark red). Asterisks represent unpaired post-hoc group differences following significant group effect in the ANOVA. (**c**) Schematic progression of early axonal damage. Figure 5c has been adapted from Figure 1E from **Luchicchi et al., 2021**.

portion of the normal-appearing white matter in MS (reported in *Figure 4—figure supplement 1*), no MRI parameter except the MRI axonal diameter proxy is significantly associated with disease duration.

## Discussion

In this work, we used a preclinical model of acute axonal damage to demonstrate that MRI-based axonal diameter mapping is sensitive to axonal degeneration. We then applied the same MRI preclinical protocol to a cohort of patients with MS and age-matched healthy controls, uncovering diffuse axonal damage in the NAWM of MS patients that was inversely associated with disease duration.

Neuropathologically, early axonal damage in MS manifests through the formation of varicosities and spheroids that enlarge the axonal diameter and are associated with impaired axonal transport (*Criste et al., 2014*). Accordingly, histological postmortem (*Bergers et al., 2002*) and animal studies (*Nikić et al., 2011*) report an increase in the mean axonal diameter in MS compared to controls in demyelinating and non-demyelinating areas. This increase is also influenced by the higher vulnerability of smaller axons compared to larger ones (*Tallantyre et al., 2010*), which implies that smaller axons are lost earlier. Histological alterations in axonal morphology have also been observed in the

NAWM of MS brain specimens in the absence of inflammation (*Luchicchi et al., 2021*), suggesting that an imbalance of axon-myelin units could represent the primary event in MS pathogenesis.

Based on neuropathological evidence, axonal damage in MS would reasonably manifest as an increase in axonal caliber measured with imaging. An increase in axonal diameter in MS was indeed reported in previous MRI studies (*Huang et al., 2016*), but the results were not validated. In the context of multicompartment models for diffusion signals, comparing imaging results with pathological evidence is fundamental to validate the model, as multicompartment models make numerous assumptions and simplifications (to cite a few: fixed diffusivities [*Alexander et al., 2010*], no exchange [*Lasič et al., 2011*], indirect account of the volume occupied by myelin [*Assaf et al., 2008*]).

Previous studies reported correlations between the axonal diameter measured with MRI and axonal caliber estimated using electron microscopy in healthy tissue (*Barazany et al., 2009*), but with a reduced sample size. In addition, none thus far have demonstrated that AxCaliber is sensitive to axonal damage. Quantitative comparison between MRI maps and stained sections is severely hampered by the fixation process and other limitations (*Horowitz et al., 2015*; *Barakovic et al., 2023*); here, we used a well-characterized rodent model in which the axonal compartment is selectively damaged. This approach has been previously used to prove the capability of diffusion MRI to dissect astrocyte and microglia reactivity in gray matter (*Garcia-Hernandez et al., 2022*). Here, we detected increased neurofilament staining intensity in the damaged tract, demonstrating altered axonal morphology without alteration in the total amount of myelin. This change is picked up by MRI as an increase in the MRI axonal diameter proxy. Electron microscopy and axonal diameter quantification demonstrate that the increase observed in imaging reflects true morphological axonal alterations, hallmarks of axonal pathology. The significant correlation between neurofilament staining intensity and the MRI axonal diameter proxy further validates the imaging parameter as a marker of axonal damage, and is compatible with recent work studying the association of MRI metrics sensitive to axonal pathology and the serum neurofilament light chain biomarker in humans (*Rahmanzadeh et al., 2021*). Overall, our preclinical results show that axonal diameter mapping through MRI can detect axonal pathology in vivo and can thus be used as a biomarker of axonal damage in MS and possibly other neurological diseases. These results are extremely important given the ongoing debate on the discrepancy between MRI axonal diameter proxy and axonal diameter measured in fixated tissue using electron microscopy. The fact that both MRI and STEM detect a significant increase in the axonal diameter reconciles such findings, demonstrating that while absolute quantification of axonal caliber might differ (with MRI consistently overestimating axonal diameter due to higher weightings of larger axons, and STEM consistently underestimating axonal diameter due to heavy tissue fixation, as shown by our shrinkage analysis), both are sensitive to the same underlying pathological phenomenon. In many clinical contexts, detecting differences in pathological versus healthy conditions is far more interesting and valuable than absolute quantification, and we firmly believe this result validates the use of MRI-based axonal diameter mapping. In addition, we quantified the shrinkage due to the sample processing as high as 39% in our preparation. While this value cannot be directly used to correct STEM quantification, due to possible different shrinkage of extra- and intracellular space (*Barakovic et al., 2023*), it partially explains the discrepancy between STEM and MRI axonal diameter proxy, validating even further the used framework. Since this is an often-disregarded issue, we believe that this piece of information would help future validation of MRI-based microstructural characterization.

Our work used an AxCaliber formulation that allows accounting for multiple fibers and thus accesses whole-brain axonal diameter mapping. Indeed, while previous findings focused on the corpus callosum (*Huang et al., 2016*), our results indicate an increase in MRI axon diameter proxy in MS patients in all the major tracts, demonstrating diffuse axonal pathology in the NAWM. Notably, the combination of whole brain mapping and a larger sample size uncovered significant negative associations between MRI axonal diameter proxy and disease duration, suggesting enlarged axonal caliber as an early event in MS pathogenesis, as illustrated in *Figure 5c*. Other tested MRI parameters are not associated with disease duration. This result indicates that consolidated axonal loss can manifest as a reduction in axonal diameter, meaning that the increase in the MRI axonal diameter proxy observed in our MS population is a candidate marker for acute axonal damage. The presence of axonal pathology so early in the disease is an incredibly meaningful result that puts the spot of MS pathogenesis in degeneration rather than myelin loss (*Luchicchi et al., 2021*).

The lack of significant correlations with clinical variables (EDSS and SDMT) can possibly be explained by a lack of power due to limited sample size or potentially compensatory phenomena in our MS population, mainly composed of patients with relatively low disease scores. Although out of the scope of this work, it would be interesting to assess whether the MRI axonal diameter proxy is associated with more relevant clinical measures like the disability progression independent of relapse activity (*Granziera et al., 2023*).

This study has some limitations. Despite some inevitable minor differences due to different brain sizes and magnet features, the human protocol was built to match the main characteristics of the preclinical diffusion sequence, such as the b-value and diffusion time range. The chosen b-value has been a compromise between sensitivity to small structures and the signal-to-noise ratio (SNR) achievable in vivo and in the MS population, as indicated by recent animal (*Crater et al., 2022*) and human (*Jensen et al., 2016*; *McKinnon et al., 2017*; *Moss et al., 2019*) work, pointing at 3000–4000 s/mm$^2$ as the b-value for which the intra-axonal water signal starts to be observable. While some works question this value as too low to detect intra-axonal signal (*Veraart et al., 2020*), our shrinkage analysis suggests that current estimated sizes fed into the calculation to determine the minimum b-value needed to measure axonal diameter through MRI, as the one reported in *De Santis et al., 2016*, should be updated. However, while feasible in vivo and in patients, this modest b-value is still suboptimal to measure axonal diameter, and higher b-values can boost the sensitivity to the intra-axonal compartment (*Veraart et al., 2020*; *Barakovic et al., 2023*). In this context, including spherical mean techniques might improve axonal diameter and intra-axonal diffusivity estimations by factoring out the effect of fibre distribution, and by providing better SNR (*Fan et al., 2020*; *Veraart et al., 2020*; *Dhital et al., 2019*).

In a subset of animals, we tested a protocol with a lower b-value and a simpler AxCaliber model (without diffusion time dependency), with the aim of facilitating future clinical AxCaliber studies. We found no qualitative differences in the outcome: MRI axonal diameter proxy was increased following fimbria damage. This result should be taken with caution due to possible predominance of extra-axonal signal as a confounding factor (*Burcaw et al., 2015*); thus, further work and perhaps more realistic simulations, considering real cell composition and morphology, are needed to clarify this issue.

We must emphasize that our quantification of the axonal diameter distribution is conducted through a Poisson distribution, which utilizes a single parameter to define both the average and dispersion. Consequently, our current methodology does not permit us to ascertain whether the observed increase is attributable to a preferential loss of small caliber axons or axonal swelling. Future investigations employing a more comprehensive experimental protocol and higher b-value, possibly using non-parametric estimates as done by *Romascano et al., 2020*, may provide the opportunity to measure the full diameter distribution, thereby offering more detailed insights into the underlying pathology.

In addition, the AxCaliber model does not take into account fiber dispersion present in the white matter. While some work point at modest values of dispersion at least in single-fiber areas (*Mollink et al., 2017*), other measured values as high as 20 degrees (*Ronen et al., 2014*; *Lee et al., 2019*). While it is reasonable to expect that some axonal directional dispersion in areas of a single fiber is likely accounted for by the second fiber population, possibly mitigating the effect, future work should include spherical mean techniques to fully remove this bias.

It is important to stress that the aim of this work is not to propose a new animal model of MS, a disease that only affects humans, but rather to validate axonal damage detection (independently from the pathology that has induced it) through noninvasive MRI, and apply the framework to characterize axonal pathology in MS. The same innovative validation framework proposed here can be used in the future to dissect the sensitivity of other more refined methods to detect axonal pathology.

We cannot exclude that other pathological processes including glial activation and gliosis could have contributed, at least in part, to the observed changes in diffusion measures. However, in contrast to our findings of axonal swelling, these processes tend to increase with advancing of disease duration and stage (*Gallego-Delgado et al., 2020*). Additionally, pathological glial changes in MS do not occur in isolation but often in association with axonal changes. Future diffusion models combining estimates of changes either in soma or fiber diameter and density could help further elucidate this aspect.

Last, the method has been demonstrated preclinically only looking at one specific tract (the fimbria, as the tract with the largest number of hippocampal projections in the rat); however, it would be

straightforward to extrapolate our results to other tracts. Future studies are needed to demonstrate this.

In conclusion, given the central role of axonal pathology in MS, developing and validating strategies for early detection of axonal damage, in vivo and noninvasively, is of high priority; our results have the potential to improve early detection and monitoring of axonal pathology in the disease, as well as provide a novel imaging marker for monitoring the effects of treatments on the progression of axonal degeneration.

## Materials and methods
### Animal preparation
Animal preparation (n=19 rodents) was carried out as described before (*Garcia-Hernandez et al., 2022*). Briefly, axonal damage in the fimbria was achieved by injecting 1 μl of ibotenic acid (a selective agonist of N-methyl-D-aspartate (NMDA) glutamate receptors that produces selective neurotoxicity *Zinkand et al., 1992*) at a concentration of 2.5 μg/μl in the dorsal hippocampus (coordinates bregma –3.8 mm, sup-inf 3.0 mm, 2 mm from the midline in the left hemisphere) (*Figure 1a*). Each animal was used as its own control by injecting the same amount of saline in the opposite hemisphere. The injection does not infect the contralateral structure, as previously reported (*Garcia-Hernandez et al., 2022*). Neuronal degeneration in the hippocampus translates into axonal loss in its major axonal output bundle, the fimbria, which is therefore used as a model for Wallerian-like axonal degeneration (*Conforti et al., 2014*). Fourteen days after surgery, rats underwent MRI scans in vivo using the AxCaliber protocol and were immediately perfused. N=9 animals were processed for immunohistological analysis, while n=6 animals were prepared for electron microscopy pipeline. Histological analysis was used to stain neuronal somas (NeuN) and quantify neuronal death in the hippocampus, neurofilaments and MBP to quantify axonal integrity and myelination in the fimbria, respectively. Electron microscopy was used to quantify axonal diameter and count in the fimbriae. N=3 additional animals were used to measure brain shrinkage from the in vivo condition to post-perfusion, post-fixation and post- sample embedding.

### Subjects
The local institutional review board approved this study and written informed consent was obtained from all participants. Eleven MS patients (age range 26–57, 6 males) and ten healthy controls (age range 23–53, 4 males) participated in the study. The minimum sample size needed to detect the effect was calculated based on previous literature (*Huang et al., 2016*). Age and sex were matched across groups. Eligibility criteria in patients were a diagnosis of relapsing-remitting MS (*Polman et al., 2011*), being on stable disease-modifying treatment or no treatment for at least 3 months, absence of clinical relapse within 3 months, and absence of corticosteroid use within one month from study enrollment. A neurologist assessed physical disability according to the EDSS (*Kurtzke, 1983*) and cognitive ability using the SDMT. Demographic and clinical data are shown in *Table 1*.

**Table 1.** Demographic characteristics of the studied cohort, including age/sex, disease duration, EDSS, SDMT, and MS treatment.
The reported p-value is the outcome of the chi-square test comparing MS and healthy controls.

| | HC (n=10) | | MS (n=11) | | p value |
|---|---|---|---|---|---|
| Age (mean and SD) | 35 y | +/-11 y | 43 y | +/-12 y | 0.27 |
| Sex | 6 M | | 4 M | | 0.13 |
| Disease duration (mean and SD) | - | | 6.40 | +/-5.47 | |
| EDSS (median, min/max) | - | | 2 | 1/4.5 | |
| SDMT (mean z score and SD) | - | | –0.70 | +/-1.47 | |
| Medication | - | | 1 avonex; 1 plegridy; 2 tecfidera; 1 gilenya; 3 ocrelizumab; 2 copaxone; 1 Rituximab | | |

## MRI acquisition

### Rats

MRI was performed on a 7T scanner (Bruker, BioSpect 70/30, Ettlingen, Germany) featuring a maximum gradient intensity of 700 mT/m. Diffusion Weighted Magnetic Resonance Imaging (DW-MRI) data were acquired using a stimulated echo planar imaging diffusion sequence, with 132 uniform distributed gradient directions, b=0 (3), 2000(15) and 4000(15) s/mm$^2$, diffusion times (Δ) 15, 25, 40 and 60ms, diffusion pulse width 5ms, diffusion duration of 5ms, repetition time (TR)=7000ms and echo time (TE)=25ms. Fourteen slices were set up centered in the fimbria with field of view (FOV)=25 × 25 mm$^2$, matrix size = 110 × 110, in-plane resolution = 0.225 × 0.225 mm$^2$ and slice thickness = 0.6 mm. The total acquisition time was 1 hr. A subset of nine animals underwent a similar protocol, with slightly lower b-values (1000 and 2500 s/mm$^2$) to explore a protocol with better clinical compatibility. Finally, three animals underwent a T2-weighted high resolution MRI protocol with full brain coverage to measure the brain volume in vivo. The T2-weighted sequence was acquired using a Rapid Acquisition with Relaxation Enhancement sequence with TR = 6253ms, TE = 11ms, 4 averages. Fifty-six slices covered the whole brain with field of view 25×25 mm$^2$, matrix size 200×200, in-plane resolution 0.125×0.125 mm$^2$ and slice thickness 0.5 mm.

### Humans

All participants were scanned on a Siemens 3T Connectom scanner, a customized 3T MAGNETOM Skyra system (Siemens Healthcare, Erlangen, Germany) housed at the MGH/HST Athinoula A. Martinos Center for Biomedical Imaging, Boston, Massachusetts, USA. The Connectom scanner is equipped with gradient coils capable of generating a maximum gradient strength of 300 mT/m, hence allowing minimization of δ (gradient duration) and echo times even at high b-values. A 64-channel brain array coil (**Keil et al., 2013**) was used for data acquisition. DW-MRI data were acquired using a spin echo echo planar imaging diffusion sequence, with 273 uniformly distributed gradient directions, b=0 (1), 2000(30), and 4000(60) s/mm$^2$, diffusion times (Δ) 17, 35, and 61 ms with four nondiffusion weighted images, diffusion pulse width 7ms, TR = 5000 ms and TE = 89ms. Eighty-two slices were set up to cover the whole brain with FOV = 220 × 220 mm$^2$, matrix size = 110 × 110, in-plane resolution = 2 × 2 mm$^2$ and slice thickness = 2 mm, partial Fourier factor 7/8, GRAPPA acceleration factor 2. In addition, anatomical images were acquired using 3D sequences with a 1.0 mm isotropic voxel size: $T_1$-weighted multiecho magnetization-prepared rapid gradient-echo images were acquired in all participants (**van der Kouwe et al., 2008**). Fluid-attenuation inversion recovery (FLAIR) images were also acquired in MS patients for white matter lesion segmentation. The total acquisition time was around 1 hr.

## Tissue processing for immunohistochemistry

Rats were deeply anesthetized with a lethal dose of sodium pentobarbital, 46 mg/kg, injected intraperitoneally (Dolethal, E.V.S.A. laboratories., Madrid, España). Rats were then perfused intracardially with 100 ml of 0.9% phosphate saline buffer (PBS) and 100 ml of ice-cold 4% paraformaldehyde (PFA, BDH, Prolabo, VWR International, Louvain, Belgium). Then, brains were immediately extracted from the skull and fixed for 1 hr in 4% PFA. Afterwards, brains were included in 3% agarose/PBS (Sigma–Aldrich, Madrid, Spain) and cut in a vibratome (VT 1000 S, Leica, Wetzlar, Germany) into 50-μm-thick serial coronal sections.

Coronal sections were rinsed and permeabilized three times in 1 x PBS with Triton X-100 at 0.5% (Sigma–Aldrich, Madrid, Spain) for 10 min each and then blocked in the same solution with 4% bovine serum albumin (Sigma-Aldrich, Madrid, Spain) and 2% goat serum donor herd (Sigma-Aldrich) for 2 hr at room temperature. The slices were then incubated overnight at 4 °C with primary antibodies against myelin basic protein (1:250 Millipore Cat# MAB384-1ML, RRID:AB_240837), neurofilament 160 kD medium (1:250, Abcam Cat# ab134458, RRID:AB_2860025) and NeuN (1:250, Millipore Cat# MAB377, RRID:AB_2298772) to label myelin, axonal processes and nuclei, respectively. The sections were subsequently incubated in specific secondary antibodies conjugated to the fluorescent probes, each at 1:500 (Molecular Probes Cat# A-11029, RRID:AB_2534088; Molecular Probes Cat# A-11042, RRID:AB_2534099) for 2 hr at room temperature. Sections were then treated with 4′,6-Diamidine-2′-phenylindole dihydrochloride at 15 mM (DAPI, Sigma-Aldrich) for 15 min at room temperature. Finally, sections were mounted on slides and covered with an anti-fading medium using a mix solution 1:10 Propyl-gallate: Mowiol (P3130, Sigma-Aldrich; 475904, MERCK-Millipore,

Massachusetts, United States). For myelin labeling, antigen retrieval was performed in 1% citrate buffer (Sigma-Aldrich) and 0.05% Tween 20 (Sigma-Aldrich) warmed to 80 °C for protein unmasking.

The tissue sections were then examined using a computer-assisted morphometry system consisting of a Leica DM4000 fluorescence microscope equipped with a QICAM Qimaging camera 22577 (Biocompare, San Francisco, USA) and Neurolucida morphometric software (MBF, Biosciences, VT, USA). Myelin, neurofilament and neural nuclei fluorescent analysis was performed using Icy software (*de Chaumont et al., 2012*). For neural nuclei, two ROIs of 200 µm² were placed per hippocampus per hemisphere in at least 5 slices per rat to obtain the corresponding intensity values. Similarly, for MBP and neurofilaments, an ROI of 400 µm² was placed per fimbria per hemisphere in at least 5 slices per rat.

## Tissue processing for electron microscopy

Rats were deeply anesthetized with a lethal dose of sodium pentobarbital, 46 mg/kg, injected intra-peritoneally (Dolethal, E.V.S.A. laboratories., Madrid, Spain). Afterwards, rats were transcardially perfused with 100 ml of 0.9% PBS and 100 ml of a fixative solution containing 2% paraformaldehyde and 2.5% glutaraldehyde in 0.1 M cacodylate buffer (pH 7.3) (Electron Microscopy Science, USA). The brains were quickly removed and postfixed overnight in the same fixative solution at 4 °C. The following day, the brains were washed with 0.1 M cacodylate buffer. Subsequently, the fixed brains were sliced into 250-µm-thick horizontal sections using a vibratome (Leica VT1000S, Germany). The sections were collected in cacodylate buffer, and those containing the fimbria were washed three times with 0.1 M cacodylate buffer for 15 minutes each, and subsequently postfixed with 1% osmium tetroxide in 0.1 M cacodylate buffer for 1.5 hr at 4 °C. The tissue was then washed in distilled water twice for 15 min each and dehydrated in a graded series of ethanol solutions, followed by propylene oxide. The sections were then infiltrated with a mixture of propylene oxide and Agar 100 embedding resin (Agar Scientific, UK) for 2 hr at room temperature, and then placed in fresh embedding resin overnight at room temperature. The following day, the samples were transferred to fresh embedding resin and polymerized for 30 hr at 60 °C in flat silicon moulds. Ultrathin sections (90 nm) were cut using an ultramicrotome (Leica UC7) and placed on formvar-coated copper slot grids. The ultrathin sections were then stained with lead citrate and imaged using STEM on a scanning electron microscope (Zeiss GeminiSEM 460, Germany).

Photos were binarized, and axons were quantified semiautomatically by two operators blind to animals and conditions (ACC and SDS). While the cell's inner area is detected automatically using the MATLAB function *bwconncomp*, nonaxonal structures are eliminated via visual screening. Six photos per condition per animal were analyzed, generating a total segmented number of axons of 12272. Axonal diameter and count were compared across conditions using an unpaired t test.

## Volume measurements

Volume measurements were taken using high-resolution T2-weighted images for in vivo conditions. Four different measurements were obtained for ex vivo conditions: immediately after perfusion, and at 4-, 7-, and 10 days post-perfusion. The volumes were extracted as follows: in vivo volumes were calculated by counting the number of voxels corresponding to brain tissue in the high-resolution T2-weighted images, multiplied by the voxel volume. The volume of the perfused brains was measured using Archimedes' principle. Briefly, a predetermined volume of fresh fixative solution was placed in a test tube, and the fixed brain was inserted into it. The difference between the final and initial volumes was considered as the volume occupied by the brain.

## Data analysis

Paired t tests were used to assess the differences in histological quantities between injected versus control hemispheres. Diffusion-weighted rat MRI data were preprocessed as described here (*De Santis et al., 2019a*). The mean signal-to-noise (SNR) of the b0 images, calculated according to *Aja-Fernández et al., 2015* was 11.2 for rats (fimbria average) and 17.3 for humans (white matter average). We also tested an alternative method (*Koay and Basser, 2006*) for SNR calculation. While the average SNR quantification was very similar between the two approaches, in the Koay and Basser SNR maps we observed artefacts due to the iterative process, so we decided to use the method by *Aja-Fernández et al., 2015*.

Human diffusion-weighted MRI data were preprocessed with software tools in FreeSurfer V5.3.0 (https://surfer.nmr.mgh.harvard.edu) and FSL V5.0 (https://fsl.fmrib.ox.ac.uk). Preprocessing included gradient nonlinearity correction, motion correction, and eddy current correction, including corresponding b-matrix reorientation. Additional preprocessing details are available at http://www.human-connectome.org/. In both rats and humans, MRI data were employed to fit the AxCaliber model using in-house software written in MATLAB R2015b (The Mathworks) to extract the average axonal diameter proxy. The theoretical framework described here (*De Santis et al., 2019b*) was modified to include the dependency of the extra-axonal signal on the diffusion time through a linear term, in the form $D^{RADIAL}_{\Delta MIN}+slope*(D- D_{MIN})$. While this is a simplification of the proposed Δ dependency (*Burcaw et al., 2015*), we believe that it is a parsimonious choice that is supported by the relatively short range of Δ values explored as compared to previous work exploring time dependency (*De Santis et al., 2016*; *Fieremans et al., 2016*), and also by comparison of a linear with a non-linear model in our data through a Bayesian information criterion (BIC), which preferred a linear fit over the expression used in *De Santis et al., 2016* in 100% of the subjects. The comparison between the two functional forms fitted to the radial diffusivity for the dataset acquired with the lower b-value protocol is reported in *Figure 1—figure supplement 3*. The fit is implemented through a cascade model as done in *Harms et al., 2017*, so that an initial CHARMED fit (*Assaf and Basser, 2005*) is performed using the data acquired at the shorter diffusion time to initialize the volume fractions. The intra-axonal axial diffusivity and the main fiber orientations are estimated through the CHARMED fit and kept fixed in the AxCaliber fit. The radial diffusivity (at the shortest Δ) in the extra-axonal compartment is first modelled using the tortuosity approximation (*Zhang et al., 2012*), and then this constraint is released in a last iteration of model fitting where everything in the model is fixed except for the radial diffusivity and the noise factor. As such, the fitted parameters are: the restricted main orientations and fractions, the intra-axonal axial diffusivity, the extra-axonal radial diffusivity, the slope of the extra-axonal radial diffusivity decay for increasing Δ, the axonal diameter and the Rician noise term. For STEAM data, an additional T1 decay is included in the fit. The BIC preferred a mono-exponential T1 decay over a bi-exponential decay in 98% of the examined voxels. Simulations using Rician noise were run on $10^5$ different combinations of parameters sampled randomly from a uniform distribution in the following ranges: axonal diameter 0.5–5 µm, volume fraction 0.1–0.5, intra-axonal axial diffusivity 0.7–$2.2x10^{-3}$ $mm^2$/s. The simulations were repeated with a narrower range of intra-axonal axial diffusivity (1.7–$2.2x10^{-3}$ $mm^2$/s) matching more closely the scenario proposed by *Dhital et al., 2019*. Each configuration was simulated 10 times by adding different Rician random noise with two different values of SNRs, matching human and animal acquisitions. The results demonstrate excellent agreement between ground truth and fitted axonal diameter for both human and animal acquisitions: $r$=0.90 and 0.75 respectively for a single repetition, $r$=0.98 and 0.95 respectively for 10 repetitions, and $r$=0.92 and 0.79 respectively for the narrower intraaxonal axial diffusivity range. The simulations are shown in *Figure 4—figure supplement 2*.

For all MS patients, lesion masks were segmented on the FLAIR images using a semiautomated method (3D-Slicer v4.2.0; https://www.slicer.org).

In rats, the fimbria was reconstructed bilaterally using the DTI-based tractography (*Figure 1b*) algorithm in the software ExploreDTI (*Leemans et al., 2009*), which was set to employ the lowest Δ and b-value. Mean axonal diameter values along the tracts were obtained for each animal in both hemispheres. Injection sites were aligned and considered the origin of the analyzed tract portion. Paired t tests were used to assess differences in the axonal diameter between the injected and contralateral fimbria, and repeated-measure ANOVA (factors: tract location, treatment (ibotenic acid *vs.* saline) and tract location*treatment) was used to assess the effect of ibotenic acid injection. p-Values were corrected according to Greenhouse-Geisser approach when sphericity assumption was not met. Post-hoc comparisons between treatment for each tract location were corrected for multiple comparisons according to the false discovery rate approach.

For groupwise analysis of NAWM, we employed a previously detailed approach (*De Santis et al., 2019a*). Briefly, fractional anisotropy maps (calculated using the lowest Δ and b-value) were employed to initialize the first steps of an improved version of the TBSS (*Smith et al., 2006*). This version performs the coregistration steps using extremely accurate tools (*Klein et al., 2009*). The warping procedure accounts for lesion masks by excluding them from the similarity metric calculation, a permutation-based nonparametric inference approach to general linear modeling (*Winkler et al.,*

*2014*). We tested for a general linear model comprising group and disease duration as regressors. Our hypothesis, compatible with postmortem evidence (*Luchicchi et al., 2021*), is that axonal blistering is an early event in the disease. Lesion masks were excluded from the statistical analysis, and multiple comparisons across clusters were controlled for by using threshold-free cluster enhancement. In addition, we tested for voxelwise associations of axonal diameter with EDSS and SDMT in MS patients only. Lastly, we calculate the average MRI axonal diameter proxy of each subject in the whole white matter (excluding lesions in MS) and compared the average across the three groups: healthy controls, MS with less than 5 years of disease duration, and MS with 5 or more years of disease duration.

## Acknowledgements

We thank Aroa Sanz Maroto for excellent technical support and the ISABIAL electron microscopy service for assistance with experiments and quantification. We thank Dr. S Aja-Fernández for assistance with signal-to-noise calculations. Funding: This work was supported by NIH 1R21NS123419-01 to CM. SDS was supported by the the Spanish Ministerio de Ciencia e Innovación, Agencia Estatal de Investigación (PID2021-128909NA-I00), by the "Centro de Excelencia Severo Ochoa" Grant CEX2021-001165-S funded by MCIN/AEI/10.13039/501100011033, and by the Generalitat Valenciana through a Subvencion a la Excelencia de Juniors Investigadores (SEJI/2019/038) and a Subvencion para la contratación de investigadoras e investigadores doctores de excelencia 2021 (CIDEGENT/2021/015). ACC is supported by the Generalitat Valenciana through a PhD fellowship ACIF/2020/301. JAG-S is supported by a Miguel Servet Fellowship from the Spanish Health Institute Carlos III (CP22/00078).

## Additional information

### Funding

| Funder | Grant reference number | Author |
|---|---|---|
| National Institutes of Health | NIH 1R21NS123419-01 | Caterina Mainero |
| Agencia Estatal de Investigación | PID2021-128909NA-I00 | Silvia De Santis |
| Ministry of Science, Innovation and Universities | CEX2021-001165-S | Silvia De Santis Antonio Cerdán Cerdá Jose A Gomez-Sanchez |
| Generalitat Valenciana | SEJI/2019/038 | Silvia De Santis |
| Generalitat Valenciana | CIDEGENT/2021/015 | Silvia De Santis |
| Generalitat Valenciana | ACIF/2020/301 | Antonio Cerdán Cerdá |
| Instituto de Salud Carlos III | CP22/00078 | Jose A Gomez-Sanchez |

The funders had no role in study design, data collection and interpretation, or the decision to submit the work for publication.

### Author contributions

Antonio Cerdán Cerdá, Formal analysis, Investigation, Methodology, Software, Visualization, Writing – original draft; Nicola Toschi, Conceptualization, Supervision, Investigation, Methodology, Writing – original draft, Writing – review and editing; Constantina A Treaba, Valeria Barletta, Ambica Mehndiratta, Software; Elena Herranz, Software, Methodology; Jose A Gomez-Sanchez, Resources, Methodology; Caterina Mainero, Conceptualization, Supervision, Funding acquisition, Methodology, Writing – original draft, Writing – review and editing; Silvia De Santis, Conceptualization, Data curation, Formal analysis, Funding acquisition, Investigation, Methodology, Project administration, Resources, Software, Supervision, Validation, Visualization, Writing – original draft, Writing – review and editing

### Author ORCIDs

Antonio Cerdán Cerdá ⓘD https://orcid.org/0000-0001-6641-014X
Jose A Gomez-Sanchez ⓘD http://orcid.org/0000-0002-6746-1800

Silvia De Santis ⓘ https://orcid.org/0000-0001-9739-6926

### Ethics

The local institutional review board at the Massachussets General Hospital approved this study, and written informed consent was obtained from all participants.

All animal experiments were approved by the Institutional Animal Care and Use Committee of the Instituto de Neurociencias de Alicante, Alicante, Spain, and comply with the Spanish (law 32/2007) and European regulations (EU directive 86/609, EU decree 2001-486, and EU recommendation 2007/526/EC; Project MRI-STRUCTURE 677/2018). The ARRIVE 10 checklist was used.

### Decision letter and Author response

Decision letter https://doi.org/10.7554/eLife.79169.sa1
Author response https://doi.org/10.7554/eLife.79169.sa2

## Additional files

### Supplementary files

- MDAR checklist

### Data availability

The preclinical data (imaging and histology) and the human imaging data that support the findings of this study are available in the open repository DIGITAL.CSIC. The human data are anonymized according to the regulations of the institution where they were acquired (Massachusetts General Hospital, Boston, USA). The software used to process the imaging data is available in the open repository DIGITAL.CSIC.

The following datasets were generated:

| Author(s) | Year | Dataset title | Dataset URL | Database and Identifier |
|---|---|---|---|---|
| De Santis S, Cerdán Cerdá A, Antonio T, Toschi N, Mainero C | 2023 | Multi-shell multi-delta MRI data from multiple sclerosis patients and controls, and rats injected with ibotenic acid in one hemisphere of the hippocampus | https://doi.org/10.20350/digitalCSIC/15716 | DIGITAL.CSIC, 10.20350/digitalCSIC/15716 |
| De Santis S | 2022 | AxCaliber_3D | https://doi.org/10.20350/digitalCSIC/14601 | DIGITAL.CSIC, 10.20350/digitalCSIC/14601 |

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
