## [Editor Report]

This is a valuable study that aims to validate and translate an established non-invasive proxy measure of axonal diameter that is derived from magnetic resonance imaging. The results are solid, demonstrating alterations in the proxy measure in rodent models of axonal damage and patients with multiple sclerosis. The Discussion acknowledges weaknesses relating to the details of modelling and signal-to-noise ratio of the measurements. This work will be of interest to researchers studying the microstructural changes in neurodegeneration.

---

## [Decision Letter]

**Decision letter after peer review:**

[Editors’ note: the authors resubmitted a revised version of the paper for consideration. What follows is the authors’ response to the first round of review.]

Thank you for submitting the paper "A translational MRI approach to validate acute axonal damage detection in multiple sclerosis" for consideration by *eLife*. Your article has been reviewed by 3 peer reviewers, one of whom is a member of our Board of Reviewing Editors, and the evaluation has been overseen by a Senior Editor. The reviewers have opted to remain anonymous. We are sorry to say that, after consultation with the reviewers, we have decided that this work will not be considered further for publication by *eLife*.

Specifically, the reviewers were concerned that the validation is limited by not having a more direct measure of axon diameter. Moreover, the pathological mechanisms are not sufficiently well matched between the rodent model and human disease to draw strong conclusions. As such, the reviewers feel the manuscript is of value but would be better suited to a specialized imaging journal.

*Reviewer #1 (Recommendations for the authors):*

The rodent protocol used single-shot EPI, long diffusion times, long echo times, and low b-values. The intention would seem to be to match the rodent experiment to the in-vivo human protocol, but it doesn't account for alterations in properties in fixed tissue. It also doesn't take advantage of the better specificity that can be achieved on pre-clinical scanners with better hardware, which can provide cleaner measurements and improved estimates of biologically-relevant parameters. Some discussion of these choices is warranted.

The units reported in Figure 1d (mm) are wrong.

Figure 2 refers to "immunohistochemistry" but as far as I can tell, this study uses exclusively immunofluorescence.

Is it correct that the color code for saline vs IBO in Figure 3 is swapped compared to Figures1-2? If so, this is confusing.

Figure 4 caption notes that "The opposite contrast was not statistically significant", but as a reader I'd prefer to show this comprehensively by including the negative contrast (e.g., as a blue colormap) so I can see whether there are no regions passing significance. This will make the results that much more compelling.

Figure 5 d-e needs a label for the vertical axis.

Figure 5 caption: what is meant by "GLM followed by post hoc comparisons"?

Please provide references for all software packages (FreeSurfer, FSL, etc) in some form (DOIs, URLs or journal articles). What is meant by '(Alexander Leemans et al., n.d.)'?

Control in rodent experiments: could be made more clear that animals are intended as their own contols using two hemispheres. It is not clear in the 'Animal preparation' section that this is what is intended – this only becomes clear in the 'Data analysis' section. Is it well established that this is a robust control (e.g., that the other hemisphere is entirely unaffected)? Similarly, it is only in the 'Data analysis' section that it is clear that saline injection is used in the control hemisphere, whereas the 'Animal preparation' section refers to 'saline and ibotenic acid' as if it is a single injection.

*Reviewer #2 (Recommendations for the authors):*

This is an interesting paper that sets out to validate the use of AxCaliber as a non-invasive quantification of axonal diameter. This goal is of great importance, especially in the context of pathology. It seems to me, however, that they have missed an opportunity by not comparing the MRI measures with some actual axonal diameter measurement based on electron or confocal microscopy. Such measurements would significantly strengthen the paper and provide some evidence that the diffusion-derived measures reflect actual axonal swelling. It is difficult to interpret the data presented: we have evidence of neurodegeneration, which correlates with estimated axonal diameter, but it is possible that what results in increased estimated diameter might be affected by other effects, e.g., gliosis. So, while the data are very interesting, they are not conclusive, and the Authors should acknowledge that clearly, and explore alternative explanations in their discussion.

Specific comments:

1. In the abstract, the final statement should be toned down, in light of the above comments.

2. The mean axonal diameter reported for both, rats and humans, appears very large, suggesting that the modified AxCaliber is only sensitive to the rightmost tail of the distribution. Can the Authors comment on this?

3. The correlation between fluorescence intensity and axonal diameter is intriguing, but the 2 quantities measure different things. Perhaps it would be more convincing to look at the left vs right difference in both quantities – are the normalised differences also correlated?

4. For the results shown in Figure 4, please state clearly how the statistical comparison and correction for multiple comparisons were conducted.

5. On page 9, the authors state that they performed some ROI analysis – how were the ROIs picked? Were these chosen a priori or after the whole brain analysis (TBSS)?

6. Discussion: please explore alternative potential explanations than an actual axonal diameter increase. While the paper cited to support the possibility of axonal swelling or the formation of spheroids are very interesting, some of these occurrences are the consequence of axonal transaction, which can happen within MS lesions, but unlikely to occur in the NAWM. Is it possible that some kind of bias in the model leads other tissue changes to mimic axonal diameter enlargement?

*Reviewer #3 (Recommendations for the authors):*

1. To validate the dMRI protocol and model fitting, authors should (1) perform Monte Carlo simulations of diffusion in realistic substrates or (2) compare dMRI results with axon diameter based on histology, such as electron microscopy. Author should also perform noise propagation to evaluate the accuracy and precision of model fitting.

2. In addition to the use of strong diffusion weighting, it is also possible to maximize the dMRI signal sensitivity to axon diameter by tuning the gradient pulse width, rather than varying the diffusion time alone (time interval between gradient pulse pair). As suggested in [Neuman, JCP 1974], the signal decay due to intra-axonal restricted diffusion is roughly proportional to the pulse width and almost independent of the diffusion time.

3. To account for the fiber orientation dispersion, spherically averaged signals of dMRI could be used to estimate the axon diameter, as in [Veraart et al., *eLife* 2020].

4. The AxCaliber model provides not only the axon diameter estimate but also intra-cellular volume fraction. The value of the volume fraction and its correlation with the histology are valuable and should be reported as well.

5. Authors did not discuss other confounding factors of axon diameter mapping using dMRI, such as the tortuous/undulating axonal shape and the diffusivity time-dependence in extra-cellular space. Ignoring these factors could lead to overestimation of axon size. It is indeed difficult to design a model to accommodate all these factors, but they should be discussed.

6. Some sequence parameters of dMRI are missing, such as the gradient pulse width, partial Fourier factor, spin echo or stimulated echo, maximal gradient strength in animal scan, and acceleration factor of parallel imaging and simultaneous multi-slice if needed.

[Editors’ note: further revisions were suggested prior to acceptance, as described below.]

Thank you for resubmitting your work entitled "A translational MRI approach to validate acute axonal damage detection in multiple sclerosis" for further consideration by *eLife*. Your revised article has been evaluated by Timothy Behrens (Senior Editor) and a Reviewing Editor.

The manuscript has been improved but there are some remaining issues that need to be addressed, as outlined below:

For this study, the experimental data and findings in animals and humans are valuable. However, Reviewer 3 notes that the interpretation and model fitting are problematic.

i) Given that the myelinated axon diameter is < 1.5 micron in histology, the observed diffusion signal time-dependence (~20%) cannot be explained by restricted diffusion in thin axons (~0.1%). Extra-cellular diffusivity time-dependence could be a more reasonable interpretation.

ii) In the response 3.5, the SNR for human and rat scans is 13.2 and 7.3 in b0 image. It is surprisingly low and hard to believe that it is possible to estimate axon size under such low SNR. Their noise propagation also shows the problem.

For the paper to be publishable, the authors will need to reconsider the interpretation and the fitted model.

*Reviewer #3 (Recommendations for the authors):*

The authors performed time-dependent diffusion MR and histology in the animal model of multiple sclerosis (MS) to demonstrate the correlation of MR estimated axon diameter index and histological findings. Further, they applied the same technique to MS patients and observed the axon size increase in the early event of MS. This study is of interest for researchers studying the microstructural changes in MS and neurodegeneration. The observation of correlation between histology and the diffusion time-dependence is exciting; however, the optimization of the MRI protocol design, the implementation of model fitting, and the interpretation of the observed diffusion time-dependence are problematic. The acquisition protocol of diffusion MR is not optimal for the axon diameter estimation; in fact, it is probably more sensitive to the extra-cellular MR signal contrast. The assumptions in the proposed diffusion MR model were not detailed in this study and even in their previous studies, and the signal-to-noise ratio in the data was not high enough for a reliable fitting. The observed diffusion time-dependence could be interpreted as the result of beadings along axons, packing geometry in extra-cellular space, and the water exchange due to T1 weighting alterations with mixing times of the stimulated echo sequence in animal scans. Given that the absolute value of axon size estimation was different in histology (0.7 micron) and diffusion MR (4 micron) by a factor of ~6, the interpretation of diffusion time-dependence as a result of restricted diffusion inside axons was questionable.

1. The major concern is that the sequence parameter in diffusion MR is not optimal to estimate the axon diameter. At low b-value < = 4000 s/mm2, it is quite impossible to have enough signal decay due to restricted diffusion inside axons [Burcaw et al., NeuroImage 0215], let alone detecting the signal time-dependence in intra-cellular space. In fact, at low b-value, most of the signal decay and signal time-dependence is contributed by the hindered diffusion in extra-cellular space [Fieremans et al., NeuroImage 2016]. Furthermore, the intra-cellular signal time-dependence largely depends on the pulse width, not the diffusion time [Neuman, JCP 1974]. Authors tried to cite the study of Axon Spectrum Imaging [Gast et al., Neuroinformatics 2023] to support their protocol design of varying diffusion time at low b-value. However, Gast et al. did not apply the AxCaliber model (diffusion narrowing regime) to estimate axon size. Actually, Gast et al. applied the assumption of intra-cellular diffusion in narrow pulse limit, and neglected the diffusion time-dependence in extra-cellular space, which has been shown as the dominant source of diffusion time-dependence at low b-values. This problem has been well recognized and further solved in previous studies [Veraart et al., *eLife* 2020; Fan et al., NeuroImage 2020], where the high b-value data (b-value > 15,000 s/mm2) were included for the model fitting.

2. Authors mentioned that, at low b-value, intra-axonal "signal" dominates. However, at low b-value, extra-axonal "signal decay/signal contrast" and "signal time-dependence" dominate (Figure 4 in [De Santis et al., 2016]). A simple calculation could indicate whether authors may misinterpret the extra-cellular signal time-dependence as intra-cellular one. For the in vivo rat MR protocol, the signal time-dependence of intra-axonal model is 20.3% for an axon diameter = 4 micron (Figure 1e) and intrinsic diffusivity = 3 um2/ms using Neuman's model, and the signal time-dependence of extra-axonal model is 19.5% for a strength of time-dependence = 0.5 micron^2^ using a log(time)/time model [Burcaw et al., 2015].

3. It is required to apply many assumptions to fit the modified AxCaliber model [De Santis et al., 2016b] to diffusion data with only two b-values at three diffusion times. For example, did authors fix the value of axial diffusivity in intra- and extra-axonal space? Did authors apply the tortuosity relation in extra-cellular space to reduce the number of parameters [Zhang et al., NeuroImage 2010]? What is the value of intrinsic diffusivity in intra-cellular space? How many parameters were fitted exactly in each voxel with 1-3 fiber tracts? These assumptions were not explained even in the previous study [De Santis 2016b].

4. The applicability of the model fitting at a typical SNR (~20) on Connectome scanner should be tested by the noise propagation. In the previous revision 3.5, the noise propagation was tested at only 4 diameter values, and the mean value of 10^4^ repetitions matched the ground truth value. However, the mean value of 10^4^ repetitions for each diameter value is the result of SNR = 20*sqrt(10^4^) = 2000. The wide histogram of fitting result actually indicates the low precision in the model fitting. To perform the noise propagation, authors should apply many different parameter combinations (for example, 10^5^) with diameter, volume fraction, and extra-cellular diffusivity varied in wide ranges.

5. The modified AxCaliber model was validated by the in vivo rat brain scan, where the stimulated echo sequence was applied. However, the diffusion signal measured by stimulated echo had varying T1-weighting due to the varying diffusion time and mixing time. If the non-diffusion weighted signal (b0 signal) was mono-exponential decay with the mixing time, this T1-weighting could be canceled out via dividing DW signals by b0 signal. If the b0 signal was not mono-exponential decay, the T1-weighting variation between multiple compartments (e.g., water around myelin and water away from myelin) could lead to spurious "diffusion" time-dependence that was related with T1-weighting and exchange. It is essential to confirm that the b0 signal is mono-exponential decay with the mixing time in white matter, where the b0 signal decay is usually bi-exponential with the mixing time.

[Editors’ note: further revisions were suggested prior to acceptance, as described below.]

Thank you for resubmitting your work entitled "A translational MRI approach to validate acute axonal damage detection in multiple sclerosis" for further consideration by *eLife*. Your revised article has been evaluated by Timothy Behrens (Senior Editor) and a Reviewing Editor.

The manuscript has been improved but there are some remaining issues that need to be addressed, as outlined below:

It is clear that the review process is converging. As you'll see, Reviewer 3 requests a few further clarifications of the study alongside some requests for limitations to be covered in the Discussion. If you are able to make changes for each of these requested comments, then we will hopefully be able to handle these as editors without the need to return to reviewers.

*Reviewer #3 (Recommendations for the authors):*

The authors in general did a great job to improve the manuscript. However, some important information about the model and results was only mentioned in the revision but not shown in detail. It is essential to include these details in either Methods or supplementary materials to support their arguments in this study. More specific concerns are given below:

1. Comment R3.1: Authors performed a simple CHARMED experiment and got the value of intra-cellular axial diffusivity close to 0.7-1 um2/ms, different from the values 2.25 um2/ms in previous studies (Dhital et al., NeuroImage 2019, 189:543 and more). This is because the inclusion of multiple highly aligned fiber bundles only factors out the fiber crossing, but not the angular dispersion in each fiber bundle. This dispersion is non-trivial even in highly aligned white matter, such as corpus callosum (~ 20-25 degree dispersion in Ronen et al., BSAF 2014, 219:1773 and Lee et al., BASF 2019, 224:1469). This is the reason why people start to use the spherical mean signal to factor out both fiber crossing and fiber dispersion for the axonal diameter mapping. This should be included in the limitation.

2. Comment R3.2: For the time-dependence of extra-cellular radial diffusivity, authors used a linear time-dependent model and suggested that the linear time-dependent model has better goodness of fit than the well-known [log(Δ/δ)+3/2]/(Δ-δ/3) model in previous studies (Burcaw et al., NeuroImage 2015 and more). What is the functional form of the linear model? Is it 1/Δ? Does it really matter to use the spurious linear model, instead of the validated log(t)/t model? For the time dependence on page 20, does the δ indicate the pulse width or the diffusion time (inter-pulse duration)? In addition, the authors did not show any results related to the fit parameters of this extra-cellular linear model for time-dependence.

3. Comment R3.4: Authors showed the noise propagation of their axonal diameter model at the SNR of 17.3 (human) and 11.2 (animal). It shows that the resolution limit of the smallest detectable axonal diameter is about 1.5 micron (kind of smaller than expected at the given SNR). However, each parameter combination was repeated 10 times with different Rician noise realizations at the same SNR. Why did authors repeat it 10 times with different Rician noise realizations? Were the 10 fitting results averaged in any way (mean, median, or other selection method)? It sounds like the SNR is boosted by a factor of sqrt(10).

4. Comment R3.4 (continued): The fitting algorithm is a little convoluted and difficult to understand now. I strongly suggest sharing the code of the model fitting and noise propagation online on a public repository.

5. Comment E.2: The relation of true signal A and Rician-biased signal E(M) was explained in Koay and Basser JMR 2006, where Equation 13 shows {E(M)}^2^ ~ = A^2^ + σ_Rician^2^ at high SNR. The σ_Rician^2^ does not have a factor of 2. In the response of comment E.2, authors used the equation for E(M^2^). However, the magnitude signal is E(M), not sqrt{E(M^2^)}. Therefore, authors should cite and use the relations in Koay and Basser JMR 2006 to correct and estimate the SNR, though it may not affect the numerical results significantly.

---

## [Author Response]

[Editors’ note: The authors appealed the original decision. What follows is the authors’ response to the first round of review.]

Reviewer #1 (Recommendations for the authors):The rodent protocol used single-shot EPI, long diffusion times, long echo times, and low b-values. The intention would seem to be to match the rodent experiment to the in-vivo human protocol, but it doesn't account for alterations in properties in fixed tissue. It also doesn't take advantage of the better specificity that can be achieved on pre-clinical scanners with better hardware, which can provide cleaner measurements and improved estimates of biologically-relevant parameters. Some discussion of these choices is warranted.

We thank the reviewer for this comment. While we agree that there was room to boost the performance of the preclinical scanner, we chose to match (as well as possible, despite inevitable minor differences) the two experimental paradigms (clinical and preclinical) to support the idea that if the MRI approach can detect axonal pathology in rats, where validation is available, then we can reasonably infer that it can also detect it in humans. Importantly, we would like to stress that the rodent data are also acquired in vivo, and no fixation is performed; therefore, there is no need to adjust the b-value to compensate for the slower dynamics in fixed tissue. We have clarified this point in the text.

Regarding our choice of b=4000 s/mm2, we would like to support it with recent bibliography. Indeed, the chosen b-value has been a compromise between sensitivity to small structures and SNR, as indicated by recent animal (Crater et al., 2022) and human (Jensen et al., 2016; McKinnon et al., 2017; Moss et al., 2019) work, pointing at 3000-4000 s/mm2 as the b-value for which the intra-axonal water signal is dominant. In addition, a paper from the laboratory that first developed the Axcaliber method recently came out (DOI: 10.1007/s12021-02309630-w) demonstrating that an MRI protocol with a maximum b-value between 3000 and 4000 s/mm2 is sufficient to capture, in vivo and in humans, various well-known aspects of axonal morphometry (e.g., the corpus callosum axon diameter variation) as well as other aspects that are less explored (e.g., axon diameter-based separation of the superior longitudinal fasciculus into segments). The same paper contains resources and further bibliography supporting that the contribution of intra-axonal water to restricted diffusion signals dominates other factors (see Online Resource 1, section A of the same paper). To challenge this recent evidence from a neurobiological perspective, we include in the supplementary material a subset of experiments in animals with lower maximum b-value (2500 s/mm2, Figure S1), where we are able to detect the same effect of increased MRI axonal diameter proxy in the injected hemisphere compared to control.

Changes in the manuscript

– Discussion, pag. 12:

“Despite some inevitable minor differences due to different brain sizes and magnet features, the human protocol was built to match the main characteristics of the preclinical diffusion sequence, such as the b-value and diffusion time range. The chosen b-value has been a compromise between sensitivity to small structures and the signal-to-noise ratio (SNR), as indicated by recent animal (Crater et al., 2022) and human (Gast et al., 2023; Jensen et al., 2016; McKinnon et al., 2017; Moss et al., 2019) work, pointing at 3000-4000 s/mm2 as the b-value for which the intra-axonal water signal is dominant. However, following recent work supporting sensitivity of diffusion-weighted MRI to axonal diameter even at lower b-values (Gast et al., 2023), we tested a protocol with a lower b-value in a subset of animals, with the aim of facilitating future clinical AxCaliber studies. We found no qualitative differences in the outcome (MRI axonal diameter proxy was increased following fimbria damage). Further work and perhaps more realistic simulations, considering real cell composition and morphology, are needed to clarify this issue.”

– Materials and methods, pag. 13:

“Fourteen days after surgery, rats underwent MRI scans in vivo using the AxCaliber protocol and were immediately perfused.”

The units reported in Figure 1d (mm) are wrong.

Thank you for spotting this. In the revised manuscript, we have represented the data as % change rather than absolute values.

Figure 2 refers to "immunohistochemistry" but as far as I can tell, this study uses exclusively immunofluorescence.

Thank you for spotting this inaccuracy; it is now fixed.

Is it correct that the color code for saline vs IBO in Figure 3 is swapped compared to Figures1-2? If so, this is confusing.

The reviewer is right, thank you for spotting the error. In the revised manuscript we have fixed it. The figure is now in Supplementary material, Figure S3.

Figure 4 caption notes that "The opposite contrast was not statistically significant", but as a reader I'd prefer to show this comprehensively by including the negative contrast (e.g., as a blue colormap) so I can see whether there are no regions passing significance. This will make the results that much more compelling.

We thank the reviewer for this comment. We have added Author response image 1, which can be added to the supplementary material if considered appropriate.

**Author response image 1. sa2fig1:** Tract-based spatial statistics showing the p-value of the group comparison testing whether MRI axonal diameter proxy is decreased in multiple sclerosis versus healthy conditions, corrected for multiple comparison across voxels. No voxels survive the p<0.05 threshold.

Figure 5 d-e needs a label for the vertical axis.Figure 5 caption: what is meant by "GLM followed by post hoc comparisons"?

Thank you for spotting these inaccuracies. To account for the reviewers’ feedback, ROI analysis has been removed from the current version of the manuscript.

Please provide references for all software packages (FreeSurfer, FSL, etc) in some form (DOIs, URLs or journal articles). What is meant by '(Alexander Leemans et al., n.d.)'?

We thank the referee for spotting these inaccuracies. We have added the missing references for all software packages as URLs. We have also added in the new version the correct citation for Alexander Leemans et al., 2009.

Control in rodent experiments: could be made more clear that animals are intended as their own contols using two hemispheres. It is not clear in the 'Animal preparation' section that this is what is intended – this only becomes clear in the 'Data analysis' section. Is it well established that this is a robust control (e.g., that the other hemisphere is entirely unaffected)? Similarly, it is only in the 'Data analysis' section that it is clear that saline injection is used in the control hemisphere, whereas the 'Animal preparation' section refers to 'saline and ibotenic acid' as if it is a single injection.

We agree with the reviewer and apologize for the lack of clarity. Now the control condition in the rodent experiment is detailed early on in the new version of the manuscript. Regarding the robustness of this control preparation in rodents, this is a design that we have carried out in previously published work (see e.g. Garcia-Hernandez et al., Science Adv 2022, DOI: 10.1126/sciadv.abq2923). In this study, we injected several neurotoxins at the same coordinates and volume using each animal as its own control by injecting saline contralaterally, testing more than 40 animals in total. After thorough immunohistochemical and MRI analysis, we found no evidence of cross-contamination in any of the animals.

Changes in the manuscript

– Material and Methods -> Animal preparation, pag. 13

“Each animal was used as its own control by injecting the same amount of saline in the opposite hemisphere. The injection does not infect the contralateral structure, as previously reported (Garcia-Hernandez et al., 2022).”

Reviewer #2 (Recommendations for the authors):This is an interesting paper that sets out to validate the use of AxCaliber as a non-invasive quantification of axonal diameter. This goal is of great importance, especially in the context of pathology. It seems to me, however, that they have missed an opportunity by not comparing the MRI measures with some actual axonal diameter measurement based on electron or confocal microscopy. Such measurements would significantly strengthen the paper and provide some evidence that the diffusion-derived measures reflect actual axonal swelling. It is difficult to interpret the data presented: we have evidence of neurodegeneration, which correlates with estimated axonal diameter, but it is possible that what results in increased estimated diameter might be affected by other effects, e.g., gliosis. So, while the data are very interesting, they are not conclusive, and the Authors should acknowledge that clearly, and explore alternative explanations in their discussion.

We thank the reviewer for this comment, which triggered an additional analysis that we believe has added further value to the paper. Using electron microscopy, we found that due to neurodegeneration, axonal damage is reflected as an increase in axon diameter (new Figure 3). We believe that these recent findings strongly support the validation of our noninvasive diffusion MRI estimates of axon diameter alterations as an early-stage hallmark of axonal pathology in MS.

The reviewer suggests gliosis as a possible additional mechanism affecting axonal diameter estimation in our preclinical model and in humans. We believe that this argument is extremely valid and have therefore discussed it in the text. Glial activation is a theme of extreme interest in our laboratory, we recently published a paper demonstrating the sensitivity of DW-MRI in gray matter to glial activation (Garcia-Hernandez et al., Science Adv 2022). We believe that this issue should be addressed properly with new experiments and models, starting by including a glia compartment in the white matter model. While this is out of the scope of the present work, we now discuss the issue of gliosis as a possible confound in the paper. In this regard, we would like to point out that in our electron microscopy data, when comparing the density of segmented axons across groups, we see no significant differences (see Figure 3 panel c, right plot), which would stand against the hypothesis of cell (other than axons) swelling.

Changes in the manuscript

– Discussion, pag. 12:

“We cannot exclude that other pathological processes including glial activation and gliosis could have contributed, at least in part, to the observed changes in diffusion measures. However, in contrast to our findings of axonal swelling, these processes tend to increase with advancing of disease duration and stage (Gallego-Delgado et al., 2020). Additionally, pathological glial changes in MS do not occur in isolation but often in association with axonal changes. Future diffusion models combining estimates of changes either in soma or fiber diameter and density could help further elucidating this aspect.”

Specific comments:1. In the abstract, the final statement should be toned down, in light of the above comments.

We thank the reviewer for this suggestion. As mentioned in the above comment, following the reviewer’s recommendation, we have carried out additional experiments that support the MRI axonal diameter proxy as a noninvasive biomarker of axonal damage under a process of neurodegeneration. In view of this fact, we have modified the final statement for fitting these new results.

Changes in the manuscript

– Abstract

“Our results demonstrate that MRI-based axonal diameter mapping is a sensitive and specific imaging biomarker that links noninvasive imaging contrasts with the underlying biological substrate, uncovering generalized axonal damage in multiple sclerosis as an early event.”

2. The mean axonal diameter reported for both, rats and humans, appears very large, suggesting that the modified AxCaliber is only sensitive to the rightmost tail of the distribution. Can the Authors comment on this?

Indeed, as the reviewer points out, there is a well-known discrepancy between the quantification of axonal diameter through MRI and electron microscopy. Absolute quantification of axonal caliber differs between methods, with MRI consistently overestimating axonal diameter – due to higher weightings of larger axons – and EM consistently underestimating axonal diameter – due to heavy tissue fixation (Horowitz et al., 2015). We believe to be a strength that our paper reconciles these differences by showing that despite the limitations of each technique, both are indeed sensitive to the same underlying pathological phenomenon. We have stressed this point in the discussion.

Changes in the manuscript

– Discussion, Pag. 11:

“The fact that both MRI and STEM detect a significant increase in the axonal diameter reconcile such findings, demonstrating that while absolute quantification of axonal caliber might differ (with MRI consistently overestimating axonal diameter due to higher weightings of larger axons, and STEM consistently underestimating axonal diameter due to heavy tissue fixation (Horowitz et al., 2015)), both are sensitive to the same underlying pathological phenomenon.”

3. The correlation between fluorescence intensity and axonal diameter is intriguing, but the 2 quantities measure different things. Perhaps it would be more convincing to look at the left vs right difference in both quantities – are the normalised differences also correlated?

We thank the reviewer for the suggestion. Please see Author response image 2, the correlation analysis for MRI axial diameter proxy and neurofilament protein intensity of the difference between conditions, hemisphere injected with saline (Kc) and hemisphere injected with ibotenic acid (Ki), following the following normalization: (Ki – Kc)/Kc. Taking the normalized differences, there was no correlation (r=0.30, p=0.46). We want to note that for this analysis, the number of points has been reduced by half, thus decreasing the power of the fit (R2=0.09), which in our view explain the lack of significant correlation.

We would prefer to leave the full analysis in the manuscript, but if the reviewer considers it appropriate to add this information, we are happy to add Author response image 2 as supplementary information.

**Author response image 2. sa2fig2:** Normalized changes in neurofilament protein intensity between hemisphere plotted against the normalized changes in MRI axonal diameter proxy. Linear regression is not significant (r=0.30, p=0.46).

4. For the results shown in Figure 4, please state clearly how the statistical comparison and correction for multiple comparisons were conducted.

We apologize for the lack of clarity. Data are corrected for multiple comparisons across voxels using a threshold-free cluster enhancement paradigm implemented in FSL. The statistical comparisons are based on a general linear model implemented in FSL (randomize). This is now discussed in the methods.

Changes in the manuscript

– Materials and methods -> Data analysis

“[…] a permutation-based nonparametric inference approach to general linear modeling (Winkler et al., 2014). We tested in a general linear model comprising group and disease duration as regressors. Our hypothesis, compatible with postmortem evidence (Luchicchi et al., 2021), is that axonal blistering is an early event in the disease. Lesion masks were excluded from the statistical analysis, and multiple comparisons across clusters were controlled for by using threshold-free cluster enhancement.”

5. On page 9, the authors state that they performed some ROI analysis – how were the ROIs picked? Were these chosen a priori or after the whole brain analysis (TBSS)?

Thank you for this comment. After reading the reviewers’ feedback, we have decided to remove the ROI analysis due to redundancy with the voxelwise approach.

6. Discussion: please explore alternative potential explanations than an actual axonal diameter increase. While the paper cited to support the possibility of axonal swelling or the formation of spheroids are very interesting, some of these occurrences are the consequence of axonal transaction, which can happen within MS lesions, but unlikely to occur in the NAWM. Is it possible that some kind of bias in the model leads other tissue changes to mimic axonal diameter enlargement?

We thank the reviewer for this insightful comment. While the new electron microscopy paradigm gives strength to the presence of axonal swelling in MS by establishing a clearer translation between preclinical and clinical imaging results, we agree on the need to discuss other potential explanations such as cell swelling due to gliosis (see also comment 2.2).

Changes in the manuscript

– Discussion, pag. 12:

“We cannot exclude that other pathological processes including glial activation and gliosis could have contributed, at least in part, to the observed changes in diffusion measures. However, in contrast to our findings of axonal swelling, these processes tend to increase with advancing of disease duration and stage (Gallego-Delgado et al., 2020). Additionally, pathological glial changes in MS do not occur in isolation but often in association with axonal changes. Future diffusion models combining estimates of changes either in soma or fiber diameter and density could help further elucidating this aspect.”

Reviewer #3 (Recommendations for the authors):1. To validate the dMRI protocol and model fitting, authors should (1) perform Monte Carlo simulations of diffusion in realistic substrates or (2) compare dMRI results with axon diameter based on histology, such as electron microscopy. Author should also perform noise propagation to evaluate the accuracy and precision of model fitting.

We thank the reviewer for this comment, which triggered an additional analysis that we believe has added further value to the paper. Using electron microscopy, we found that following ibotenic injections, axonal damage is indeed reflected as an increase in axon diameter (new Figure 3). We believe that these recent findings strongly support the validation of our noninvasive diffusion MRI estimates of axon diameter swelling.

While we would like to remark that we use a framework originally proposed and validated (in terms of accuracy/precision) by the group of Assaf when originally implemented, we agree on the importance of showing the performance of the framework in our specific setting, especially its robustness to noise. Noise is modelled through a variable (the background noise level) that is estimated as an additional free parameter in the fit, as proposed in the original papers by Assaf (2004,2005 and 2008). When calculating this parameter in our preclinical and clinical data, we obtain an average of 0.0754 +/- 0.0083 (humans, in the whole white matter) and 0.137 +/- 0.0169 (rats, in whole fimbria). Please note that this value represents the proportion of noise when the total signal amplitude is normalized to 1, thus confirming that (i) noise is a small proportion of the total signal (<8% and <14%, respectively), and (ii) clinical data, as expected, have higher SNR. We then run simulations by generating the MRI signal for varying axonal diameter in the physiological range (1-4 µm), add noise matching the same SNR measured in our preclinical and clinical data, and check that the histograms of the obtained axonal diameters (calculated over 104 noisy repetitions) correctly match the gold standard value. The results reported in Author response image 3 (panel a, clinical settings, panel b, preclinical) show good agreement between real and predicted diameters, and can be added to the supplementary material if considered appropriate.

**Author response image 3. sa2fig3:** Simulations with different noise levels (matching human and animals, respectively) and different gold standards of axonal caliber, showing good agreement between real and predicted diameter.

2. In addition to the use of strong diffusion weighting, it is also possible to maximize the dMRI signal sensitivity to axon diameter by tuning the gradient pulse width, rather than varying the diffusion time alone (time interval between gradient pulse pair). As suggested in [Neuman, JCP 1974], the signal decay due to intra-axonal restricted diffusion is roughly proportional to the pulse width and almost independent of the diffusion time.

We agree with the reviewer that other choices of protocols are possible, however, the aim of this paper is not to improve an existing strategy, but rather to explore the neurobiological basis of a controversial yet used method. In our case, the employed protocol has been defined giving priority to transferability between human and animal scanners and the use of a protocol easily implementable in clinical scanners.

However, we believe that the value of this innovative validation framework is that it can be used to dissect the sensitivity of other, more refined methods to detect axonal pathology. We have made it clear in the text.

Changes in the manuscript

– Discussion, pag. 12:

“The same innovative validation framework proposed here can be used in the future to dissect the sensitivity of other more refined methods to detect axonal pathology.”

3. To account for the fiber orientation dispersion, spherically averaged signals of dMRI could be used to estimate the axon diameter, as in [Veraart et al., eLife 2020].

Thank you for the input. It is indeed a very valuable suggestion to explore other protocols, however, since the focus of the paper is the neurobiological validation, we decided to employ a standard protocol used in previous studies and recommended by the author who introduced the AxCaliber framework (see the recent paper from Assaf´s laboratory 10.1007/s12021-02309630-w). However, the answer to the previous comment also applies here: the value of this innovative validation framework is that it can be used to dissect the sensitivity of other, more refined methods to detect axonal pathology.

4. The AxCaliber model provides not only the axon diameter estimate but also intra-cellular volume fraction. The value of the volume fraction and its correlation with the histology are valuable and should be reported as well.

Thank you for the valuable suggestion. Indeed, thanks to the reviewer's suggestion, we have extended the analysis to test the volume fraction as well, both in preclinical and clinical data. In humans, we found a trend of reduction of the restricted volume signal in MS compared to control, which however do not reach significancy after multiple comparison correction. The p-value is shown in Author response image 4. No correlations with the disease durations are present.

**Author response image 4. sa2fig4:** Tract-based spatial statistics showing the p-value of the group comparison testing whether the restricted signal fraction is decreased in multiple sclerosis versus healthy conditions,corrected for multiple comparison across voxels. No voxels survive the p<0.05 threshold.

These results are in agreement with previous work by our group (see De Santis et al. Neuroimage:clinical 2019), where with a larger cohort, we could prove widespread significant reduction of the restricted signal fraction in MS normal-appearing white matter.

Interestingly, in rats, we detected significantly reduced restricted fraction, which however do not correlate with the Nf intensity staining (p=0.53). We believe that this result supports the increased specificity of the MRI axonal diameter with respect to other (easier to obtain) measures like the restricted fraction, which motivated this study in the first place. The plots in Author response image 5 summarize this finding, and can be included in the supplementary material if deemed appropriate.

**Author response image 5. sa2fig5:** (a) Mean restricted signal fraction calculated in the ibotenic vs saline-injected fimbria reconstructed using tractography. Asterisks represent significant differences (n=9, paired t test across hemispheres, p=0.012). (b) Neurofilaments fluorescence intensity plotted against the restricted signal fraction. Linear regression is not significant (p=0.53).

5. Authors did not discuss other confounding factors of axon diameter mapping using dMRI, such as the tortuous/undulating axonal shape and the diffusivity time-dependence in extra-cellular space. Ignoring these factors could lead to overestimation of axon size. It is indeed difficult to design a model to accommodate all these factors, but they should be discussed.

We thank the reviewer for this valuable suggestion. We have included these factors as limitations as requested.

Changes in the manuscript

Discussion, Pag. 12:

“The model does not account for the diffusion time-dependence in the extracellular space (De Santis et al., 2016), which might explain part of the discrepancy between histological and MRI proxies for axonal diameter. However, our extensive validation approach supports the conclusion that the used model captures salient features of the axonal microstructure.”

Discussion, Pag. 12:

“In addition, the AxCaliber model does not take into account fiber dispersion present in the white matter (Ronen et al., 2014). However, some axonal directional dispersion in areas of a single fiber is likely accounted for by the second fiber population, possibly mitigating the effect. This needs to be further validated with simulations and 3D histology.”

6. Some sequence parameters of dMRI are missing, such as the gradient pulse width, partial Fourier factor, spin echo or stimulated echo, maximal gradient strength in animal scan, and acceleration factor of parallel imaging and simultaneous multi-slice if needed.

We apologize for the missing information. We have added the requested information to the new manuscript. No simultaneous multi-slice option was used.

Changes in the manuscript

MRI acquisition, pag. 14:

“Rats: MRI was performed on a 7 T scanner (Bruker, BioSpect 70/30, Ettlingen, Germany) featuring a maximum gradient intensity of 700 mT/m. Diffusion Weighted Magnetic Resonance Imaging (DW-MRI) data were acquired using a stimulated echo planar imaging diffusion sequence, with 132 uniform distributed gradient directions, b=0(3), 2000(15) and 4000(15) s/mm2, diffusion times (∆) 15, 25, 40 and 60 ms, diffusion duration of 5 ms, repetition time (TR) = 7000 ms and echo time (TE) = 25 ms. Fourteen slices were set up centered in the fimbria with field of view (FOV) = 25×25 mm2, matrix size = 110 × 110, in-plane resolution = 0.225×0.225 mm2 and slice thickness = 0.6 mm. The total acquisition time was 1 hour.”

MRI acquisition, pag. 15:

“DW-MRI data were acquired using a spin echo echo planar imaging diffusion sequence, with 273 uniformly distributed gradient directions, b=0(1), 2000(30) and 4000(60) s/mm2, diffusion times (∆) 17, 35 and 61 ms with four non-diffusion weighted images, diffusion pulse width 7 ms, TR = 5000 ms and TE = 89 ms. Eighty-two slices were set up to cover the whole brain with FOV = 220×220 mm2, matrix size = 110×110, in-plane resolution = 2×2 mm2 and slice thickness = 2 mm, partial Fourier factor 7/8, GRAPPA acceleration factor 2.”

[Editors’ note: what follows is the authors’ response to the second round of review.]

The manuscript has been improved but there are some remaining issues that need to be addressed, as outlined below:Reviewer #3 (Recommendations for the authors):The authors performed time-dependent diffusion MR and histology in the animal model of multiple sclerosis (MS) to demonstrate the correlation of MR estimated axon diameter index and histological findings. Further, they applied the same technique to MS patients and observed the axon size increase in the early event of MS. This study is of interest for researchers studying the microstructural changes in MS and neurodegeneration. The observation of correlation between histology and the diffusion time-dependence is exciting; however, the optimization of the MRI protocol design, the implementation of model fitting, and the interpretation of the observed diffusion time-dependence are problematic. The acquisition protocol of diffusion MR is not optimal for the axon diameter estimation; in fact, it is probably more sensitive to the extra-cellular MR signal contrast. The assumptions in the proposed diffusion MR model were not detailed in this study and even in their previous studies, and the signal-to-noise ratio in the data was not high enough for a reliable fitting. The observed diffusion time-dependence could be interpreted as the result of beadings along axons, packing geometry in extra-cellular space, and the water exchange due to T1 weighting alterations with mixing times of the stimulated echo sequence in animal scans. Given that the absolute value of axon size estimation was different in histology (0.7 micron) and diffusion MR (4 micron) by a factor of ~6, the interpretation of diffusion time-dependence as a result of restricted diffusion inside axons was questionable.

We thank the reviewer for their insightful comments, which triggered several actions on our side; in brief:

A comprehensive update of the model employed to fit the data, now incorporating the time-dependency of the extracellular space as requested;

Initiation of new experiments to investigate the discrepancy between EM- and MRI-based axonal size estimation;

Conducting new experiments to evaluate mono-exponential decay in mixing time;

Revised SNR estimation and simulations.

Please find below our response to specific points.

1. The major concern is that the sequence parameter in diffusion MR is not optimal to estimate the axon diameter. At low b-value < = 4000 s/mm2, it is quite impossible to have enough signal decay due to restricted diffusion inside axons [Burcaw et al., NeuroImage 0215], let alone detecting the signal time-dependence in intra-cellular space. In fact, at low b-value, most of the signal decay and signal time-dependence is contributed by the hindered diffusion in extra-cellular space [Fieremans et al., NeuroImage 2016]. Furthermore, the intra-cellular signal time-dependence largely depends on the pulse width, not the diffusion time [Neuman, JCP 1974]. Authors tried to cite the study of Axon Spectrum Imaging [Gast et al., Neuroinformatics 2023] to support their protocol design of varying diffusion time at low b-value. However, Gast et al. did not apply the AxCaliber model (diffusion narrowing regime) to estimate axon size. Actually, Gast et al. applied the assumption of intra-cellular diffusion in narrow pulse limit, and neglected the diffusion time-dependence in extra-cellular space, which has been shown as the dominant source of diffusion time-dependence at low b-values. This problem has been well recognized and further solved in previous studies [Veraart et al., eLife 2020; Fan et al., NeuroImage 2020], where the high b-value data (b-value > 15,000 s/mm2) were included for the model fitting.2. Authors mentioned that, at low b-value, intra-axonal "signal" dominates. However, at low b-value, extra-axonal "signal decay/signal contrast" and "signal time-dependence" dominate (Figure 4 in [De Santis et al., 2016]). A simple calculation could indicate whether authors may misinterpret the extra-cellular signal time-dependence as intra-cellular one. For the in vivo rat MR protocol, the signal time-dependence of intra-axonal model is 20.3% for an axon diameter = 4 micron (Figure 1e) and intrinsic diffusivity = 3 um2/ms using Neuman's model, and the signal time-dependence of extra-axonal model is 19.5% for a strength of time-dependence = 0.5 micron^2^ using a log(time)/time model [Burcaw et al., 2015].

We appreciate the reviewer's insightful comments and wish to address them collectively, as we perceive them to be closely interconnected. We concur that higher b-values enhance sensitivity to intra-axonal water. However, we must note that the paper by Veraart et al., though offering valuable insights into the MRI sensitivity to axonal diameter estimation, includes data with very high b-values that were (i) acquired ex-vivo in animals and at an extremely high field (16.4T); and (ii) obtained with an in-vivo human protocol of partial coverage, limited resolution, and suitable only for imaging the corpus callosum. This experimental approach, fully valid for a proof-of-concept, was unfortunately incompatible with our research question, setups, in-vivo framework, and clinical population. Consequently, we opted for what we consider a balanced compromise between b-value and feasibility. We concur with the reviewer that this situation might be partially mitigated by strategies such as the spherical mean, as implemented in Barakovic et al. 2023, and have acknowledged this as a limitation of our work.

Furthermore, we would like to emphasize that the theoretical frameworks employed to determine the lower boundaries on axonal size detection necessarily rely on assumptions about tissue composition and properties that (i) can significantly alter the conclusions drawn, and (ii) are challenging to ascertain. For instance, Figure 1 of the paper by De Santis et al. 2016 clearly illustrates that the proportions of intra and extra-axonal signals critically depend on the value of intra-axonal diffusivity. While theoretical calculations typically use a value of 2.4x10-9 mm2/s (lines e-f), a simple CHARMED experiment yields values closer to 0.7-1x109 mm2/s (lines a-b). Additionally, in this revised version, we present new experiments demonstrating that STEM size estimation is substantially biased (about 40% in our preparation; see response to comment E.1) by tissue preparation steps including fixation and embedding, with literature data estimating this bias as high as 65% (Dyrby et al. 2018). This could further align with the scenario for the contribution of intra-axonal diffusivity reported in panel b or, most likely, higher.

In the light of these new results, we believe our arguments support the conclusion that, given the current uncertainties surrounding several critical parameters used in calculations, it is challenging to dismiss the sensitivity of a modest b-value, such as the one employed in our study (and in several previous papers with positive results), to axonal diameter. In this context, our biology-driven approach to generate, validate, and detect axonal pathology contributes valuable insights to an open question which is still ongoing.

Of course, all this does not mean that higher b-values do not provide enhanced sensitivity to axonal caliber measurements, so we fully agree with the reviewer on that. To stress this, we have moderated the emphasis in the paragraph concerning the b-value, aligning it more closely with the context of our study and the reviewer's insights, and have included the choice of b-value as a limitation of our work, expanding the citations to recent papers using higher b-values (e.g., Barakovic et al. 2023).

We also agree with the reviewer that the omission of the δ dependency in the model was a significant limitation of our previous analysis. Therefore, we have heeded this valuable suggestion and re-processed all the data with an updated model that includes δ dependency, specifically through a linear term in the expression of the diffusion orthogonal to the main orientation in the hindered compartment. The rationale for selecting a linear model was its parsimonious nature, especially considering that our δ sampling, compared to that used in De Santis et al. 2016, covers a more limited range (up to 60 vs. up to 200 ms). We have calculated DTI-based radial diffusivities for varying deltas, and the Bayesian Information Criterion (BIC) supports a linear fit versus the expression used in De Santis et al. 2016 in 100% of the voxels (data can be included upon request).

Upon the inclusion of this additional term, we observed a reduction in the estimated size compared to our previous analysis, mirroring the findings in De Santis et al. 2016. This adjustment brought the estimates closer to histological values and also to recent dw-MRI estimates, although the latter was achieved with a more complex diffusion-relaxation model, as detailed in Barakovic et al. 2023. Additionally, we noted that the estimated slope is consistently negative (as anticipated) and does not exhibit significant differences between conditions (p=0.20, data can be included upon request).

Importantly, after the change in the processing, all the main results of the paper still hold, namely:

higher axonal diameter in ibotenic-injected hemisphere compared to control (p=0.021), as shown in Figure 1 of the revised manuscript;higher axonal diameter in MS as compared to controls, as shown in Figure 4 of the revised manuscript;preferential increase of axonal diameter in subjects with short disease duration, as shown in Figure 5 of the revised manuscript (NB: although the voxel-wise statistic is not significant anymore but is now a trend, with the lowest p-value equal to 0.051, the group comparison in the whole WM stays significant).

In the previous version of the paper, we cited and discussed the work by Gast et al. to substantiate the sensitivity of a b-value of 4000 to axonal diameter, albeit with a different protocol. However, we concur with the reviewer that this reference may lead to confusion due to the disparate models employed. Consequently, we have elected to remove it.

Changes in the text:

Discussion, pag. 12: “The chosen b-value has been a compromise between sensitivity to small structures and the signal-to-noise ratio (SNR) achievable in vivo and in the MS population, as indicated by recent animal (Crater et al., 2022) and human (Jensen et al., 2016; McKinnon et al., 2017; Moss et al., 2019) work, pointing at 3000-4000 s/mm2 as the b-value for which the intra-axonal water signal starts to be observable.”

Discussion, pag. 12: “While some works question this value as too low to detect intra-axonal signal (Veraart et al., 2020), our shrinkage analysis suggests that current estimated sizes fed into the calculation to determine the minimum b-value needed to measure axonal diameter through MRI, as the one reported in (De Santis et al., 2016), should be updated. However, while feasible in vivo and in patients, this modest b-value is still suboptimal to measure axonal diameter, and higher b-values can boost the sensitivity to the intra-axonal compartment (Veraart et al., 2020)(Barakovic, M, 2023). In this context, including spherical mean techniques might improve axonal diameter estimation (Fan et al., 2020).”

Discussion, pag 13: “This result should be taken with caution due to possible predominance of extra-axonal signal as a confounding factor (Burcaw et al., 2015);”

References:

Dyrby TB et al. Validation strategies for the interpretation of microstructure imaging using diffusion MRI. NeuroImage 182, 62-79 (2018)

Barakovic et al. 2023 Estimating axon radius using diffusion-relaxation MRI: calibrating a surface-based relaxation model with histology. Front Neurosci 17:1209521.

3. It is required to apply many assumptions to fit the modified AxCaliber model [De Santis et al., 2016b] to diffusion data with only two b-values at three diffusion times. For example, did authors fix the value of axial diffusivity in intra- and extra-axonal space? Did authors apply the tortuosity relation in extra-cellular space to reduce the number of parameters [Zhang et al., NeuroImage 2010]? What is the value of intrinsic diffusivity in intra-cellular space? How many parameters were fitted exactly in each voxel with 1-3 fiber tracts? These assumptions were not explained even in the previous study [De Santis 2016b].

We thank the reviewer for highlighting this lack of information. We provide more details in the revised version. In summary: (1) we do not fix diffusivities, but we estimate the intracellular diffusivity (axial diffusivity) and the fiber direction from a CHARMED experiment with fixed diffusion time; (2) we use a cascade model as in Harms et al. 2017 so that the rest of the CHARMED estimated parameters (restricted fraction, axial diffusivity) are used to initialize the AxCaliber fit; and (3) in order to implement the δ-dependency, we first use a tortuosity approximation in the extracellular space to fix the radial diffusivity for the smaller δ, but then add a last iteration of model fitting to release this constraint.

Changes in the text:

– Methods, pag. 19:

“The theoretical framework described here (De Santis et al., 2019b) was modified to include the dependency of the extra-axonal signal on the diffusion time through a linear term. While this is a simplification of the proposed δ dependency (Burcaw et al., 2015), we believe that it is a parsimonious choice that is supported by the relatively short range of δ values explored as compared to previous work exploring time dependency (De Santis et al., 2016) (Fieremans et al., 2016), and also by comparison of a linear with a non-linear model in our data through the Bayesian Information Criterion (BIC), which preferred a linear fit over the expression used in De Santis et al. 2016 in 100% of the voxels. As such, the fitted parameters are: the restricted main orientations and fractions, the intrinsic diffusivity, the extra-axonal radial diffusivity, the slope of the extra-axonal signal decay for increasing δ, the axonal diameter and the Rician noise term. For STEAM data, an additional T1 decay is included in the fit. The BIC preferred a mono-exponential T1 decay over a bi-exponential decay in 98% of the examined voxels. The fit is implemented through a cascade model as done in (Harms et al., 2017), so that an initial CHARMED fit (Assaf and Basser, 2005) is performed using the data acquired at the shorter diffusion time to initialize the volume fractions. The intrinsic diffusivity and the main fiber orientations are estimated through the CHARMED fit and kept fixed in the AxCaliber fit. The axial diffusivity (at the shortest δ) in the extra-axonal compartment is first modelled using the tortuosity approximation (Zhang et al., 2012), and then this constraint is released in a last iteration of model fitting where everything in the model is fixed except the axial diffusivity and the noise factor.”

References:

Harms RL, Fritz FJ, Tobisch A, Goebel R, Roebroeck A. Robust and fast nonlinear optimization of diffusion MRI microstructure models. Neuroimage. 2017 Jul 15;155:82-96.

4. The applicability of the model fitting at a typical SNR (~20) on Connectome scanner should be tested by the noise propagation. In the previous revision 3.5, the noise propagation was tested at only 4 diameter values, and the mean value of 10^4^ repetitions matched the ground truth value. However, the mean value of 10^4^ repetitions for each diameter value is the result of SNR = 20*sqrt(10^4^) = 2000. The wide histogram of fitting result actually indicates the low precision in the model fitting. To perform the noise propagation, authors should apply many different parameter combinations (for example, 10^5^) with diameter, volume fraction, and extra-cellular diffusivity varied in wide ranges.

We thank the reviewer for this comment. Indeed, a more comprehensive simulation approach, especially after an additional parameter was included in the model to describe the dependency of the extra-axonal signal from the diffusion time, could further demonstrate the validity of the extracted measurements. As suggested, we have now run simulations using 10^5^ different combinations of parameters sampled randomly from a uniform distribution: diameter (sampled range 0.5-5 µm), volume fraction (0.1-0.5), and extra-cellular diffusivity (0.7-2.210^-9mm2/s). Each configuration was repeated 10 times with different Rician noise. We have found excellent agreement between fitted and ground truth axonal size (r=0.90 for human SNR and 0.75 for rodent SNR).

Changes in the text:

– Methods, pag. 20:

“Simulations using Rician noise were run on 10^5^ different combinations of parameters sampled randomly from a uniform distribution in the following ranges: axonal diameter 0.5-5 µm, volume fraction 0.1-0.5, extra-cellular diffusivity 0.7-2.2x10-3mm2/s. Each configuration was simulated 10 times by adding different Rician random noise with two different values of SNRs, matching human and animal acquisitions. The results demonstrate excellent agreement between ground truth and fitted axonal diameter for both human and animal acquisitions (r=0.90 and 0.75 respectively, as shown in Figure 4—figure supplement 1).”

5. The modified AxCaliber model was validated by the in vivo rat brain scan, where the stimulated echo sequence was applied. However, the diffusion signal measured by stimulated echo had varying T1-weighting due to the varying diffusion time and mixing time. If the non-diffusion weighted signal (b0 signal) was mono-exponential decay with the mixing time, this T1-weighting could be canceled out via dividing DW signals by b0 signal. If the b0 signal was not mono-exponential decay, the T1-weighting variation between multiple compartments (e.g., water around myelin and water away from myelin) could lead to spurious "diffusion" time-dependence that was related with T1-weighting and exchange. It is essential to confirm that the b0 signal is mono-exponential decay with the mixing time in white matter, where the b0 signal decay is usually bi-exponential with the mixing time.

The reviewer is right. Indeed, we confirmed that in our data, the mono-exponential fit is preferred over the bi-exponential in the vast majority of voxels by using a Bayesian information criterion to compare the two models. In figure 6, the mean % of voxels belonging to the fimbria where the mono-exponential fit is preferred, thus yielding a lower BIC, is reported.

In addition, to make sure that this was not due to the relatively few mixing times sampled, we have acquired new data on n=3 animals by halving the sampling interval. Still, in the majority of voxels the mono-exponential fit is preferred over the bi-exponential as shown in figure 6.

Changes in the text:

Methods, pag. 19:

“For STEAM data, an additional T1 decay is included in the fit. The BIC preferred a mono-exponential T1 decay over a bi-exponential decay in 98% of the examined voxels.”

[Editors’ note: what follows is the authors’ response to the third round of review.]

The manuscript has been improved but there are some remaining issues that need to be addressed, as outlined below:It is clear that the review process is converging. As you'll see, Reviewer 3 requests a few further clarifications of the study alongside some requests for limitations to be covered in the Discussion. If you are able to make changes for each of these requested comments, then we will hopefully be able to handle these as editors without the need to return to reviewers.Reviewer #3 (Recommendations for the authors):The authors in general did a great job to improve the manuscript. However, some important information about the model and results was only mentioned in the revision but not shown in detail. It is essential to include these details in either Methods or supplementary materials to support their arguments in this study. More specific concerns are given below:1. Comment R3.1: Authors performed a simple CHARMED experiment and got the value of intra-cellular axial diffusivity close to 0.7-1 um2/ms, different from the values 2.25 um2/ms in previous studies (Dhital et al., NeuroImage 2019, 189:543 and more). This is because the inclusion of multiple highly aligned fiber bundles only factors out the fiber crossing, but not the angular dispersion in each fiber bundle. This dispersion is non-trivial even in highly aligned white matter, such as corpus callosum (~ 20-25 degree dispersion in Ronen et al., BSAF 2014, 219:1773 and Lee et al., BASF 2019, 224:1469). This is the reason why people start to use the spherical mean signal to factor out both fiber crossing and fiber dispersion for the axonal diameter mapping. This should be included in the limitation.

We thank the reviewer for highlighting the key factors affecting the measured intra-cellular diffusivity in models like CHARMED, which we utilize as an initial step for the AxCaliber fit. The insight into the limitations posed by not employing the spherical mean signal is well-received. Accordingly, we have updated our manuscript to reflect these considerations, including the relevant reference.

Moreover, for the purpose of this revision, we re-conducted our simulations with a different range of intra-cellular axial diffusivities (1.7 to 2.2 um2/ms), to match more closely the cited paper. These simulations did not show a significant difference in accuracy compared to our original results. This finding is now included in the supplementary material, Figure 4 —figure supplement 2, panels e and f.

Also, we concur that the quantification of the fiber dispersion in white matter varies significantly across different studies and techniques. Still, while some, like the ones you referenced, reports a dispersion of around 20 degrees, others such as Mollink (Neuroimage 2017) suggest a lower dispersion of approximately 10 degrees or less at least in the corpus callosum. This discrepancy underscores the ongoing debate in the scientific community regarding the precise extent of inter-bundle fiber dispersion. Our study recognizes these variances and acknowledges the limitations of even gold-standard histological techniques, as discussed in our revised manuscript. We believe that a consensus on the exact amount of fiber dispersion, factoring in these methodological limitations, is yet to be established.

Discussion:

“In this context, including spherical mean techniques might improve axonal diameter and intraaxonal diffusivity estimations by factoring out the effect of fibre distribution, and by providing better SNR (Fan et al., 2020) (Veraart et al., 2020) (Dhital et al., 2019). In addition, the AxCaliber model does not take into account fiber dispersion present in the white matter. While some work point at modest values of dispersion at least in single-fiber areas (Mollink et al., 2017), other measured values as high as 20 degrees (Ronen et al., 2014), (Lee et al., 2019). While it is reasonable to expect that some axonal directional dispersion in areas of a single fiber is likely accounted for by the second fiber population, possibly mitigating the effect, future work should include spherical mean techniques to fully remove this bias.”

Materials and methods:

“The simulations were repeated with a narrower range of intra-axonal axial diffusivity (1.72.2x10-3 mm2/s) matching more closely the scenario proposed by (Dhital et al., 2019). Each configuration was simulated 10 times by adding different Rician random noise with two different values of SNRs, matching human and animal acquisitions. The results demonstrate excellent agreement between ground truth and fitted axonal diameter for both human and animal acquisitions: r=0.90 and 0.75 respectively for a single repetition, r=0.98 and 0.95 respectively for 10 repetitions, and r=0.92 and 0.79 respectively for the narrower intra-axonal axial diffusivity range. The simulations are shown in Figure 4—figure supplement 2.”

2. Comment R3.2: For the time-dependence of extra-cellular radial diffusivity, authors used a linear time-dependent model and suggested that the linear time-dependent model has better goodness of fit than the well-known [log(Δ/δ)+3/2]/(Δ-δ/3) model in previous studies (Burcaw et al., NeuroImage 2015 and more). What is the functional form of the linear model? Is it 1/Δ? Does it really matter to use the spurious linear model, instead of the validated log(t)/t model? For the time dependence on page 20, does the δ indicate the pulse width or the diffusion time (inter-pulse duration)? In addition, the authors did not show any results related to the fit parameters of this extra-cellular linear model for time-dependence.

We thank the reviewer for highlighting the need for additional details on our model for incorporating diffusion time dependency in the extra-axonal space. We utilized the linear expression: D^RADIAL^
_∆=min(∆)_+slope*(Δ-min(Δ)), which incorporates the extra-axonal radial diffusivity from the CHARMED fit used to initialize the AxCaliber. This approach helps reduce the number of free parameters, a crucial consideration given the in vivo data collection constraints. While this functional form is preferred over log(t)/t within the limited Δ range in our study, we included a comparison between these two forms in the supplementary material (new Figure 1—figure supplement 3). Additionally, we have tested both functions with diffusivity at Δ=15 ms as a free parameter. Neither the log(t)/t nor the linear model is conclusively supported by the BIC in this scenario, indicating a mixed situation with similar BIC factors and residual sum of squares, likely attributable to the limited Δ range sampled.

We were unable to locate “δ” on page 20 of our submitted PDF, which contains references. However, we have revised the manuscript to ensure consistent use of Δ for diffusion time and δ for pulse width, thereby clarifying any ambiguity.

To address the final part of the comment, in the revised manuscript, we have included the plot of the slope, radial diffusivity, and restricted fraction in the injected versus control hemisphere (new Figure 1—figure supplement 1). While none of these parameters show significant differences across hemispheres, there is a tendency (p=0.09) for a reduced slope in the injected hemisphere. We also conducted a statistical comparison of these quantities in humans, examining both group differences between MS patients and controls and the association with disease duration. The slope of the extra-axonal radial diffusivity decay and the restricted signal fraction are significantly reduced in portions of the normal-appearing white matter in MS. However, no MRI parameter, except for the MRI axonal diameter proxy, shows a significant association with disease duration. These results are documented in new Figure 4—figure supplement 1.

Materials and methods:

“The comparison between the two functional forms fitted to the radial diffusivity for the dataset acquired with the lower b-value protocol is reported in Figure 1—figure supplement 3.”

Results:

“The other parameters extracted from the MRI analysis are not significantly different between hemispheres, although there is a tendency of reduced slope of the extra-axonal radial diffusivity decay for increasing diffusion time in the injected hemisphere (Figure 1—figure supplement 1).”

“Finally, we tested both group differences and associations with disease duration for the rest of the parameters extracted in the MRI analysis. While both the slope of the extra-axonal radial diffusivity decay for increasing diffusion time and the restricted signal fraction are significantly reduced in a portion of the normal-appearing white matter in MS (reported in Figure 4—figure supplement 1), no MRI parameter except the MRI axonal diameter proxy is significantly associated with disease duration.”

Discussion:

“Other tested MRI parameters are not associated with disease duration.”

3. Comment R3.4: Authors showed the noise propagation of their axonal diameter model at the SNR of 17.3 (human) and 11.2 (animal). It shows that the resolution limit of the smallest detectable axonal diameter is about 1.5 micron (kind of smaller than expected at the given SNR). However, each parameter combination was repeated 10 times with different Rician noise realizations at the same SNR. Why did authors repeat it 10 times with different Rician noise realizations? Were the 10 fitting results averaged in any way (mean, median, or other selection method)? It sounds like the SNR is boosted by a factor of sqrt(10).

We appreciate the reviewer's attention to this detail. The 2D histogram presented in our original figure indeed corresponds to a single repetition. However, our methodology included conducting 10 repetitions with distinct Rician noise realizations. This approach reflects the standard MRI data analysis practice where spatial information is typically averaged across neighboring voxels within the same biological structure. This averaging is intended to counterbalance the noise inherent in the measurement process, a standard in ROI-based analysis. Even in voxel-based analysis, standard analysis pipelines like TBSS utilize spatial neighbourhood information to enhance the reliability of extended signal areas. In light of this, we believe that presenting the accuracy averaged over several repetitions in simulations offers a valid comparison to real data analysis. Consequently, we have included the 2D histograms for both single and 10 repetitions in the updated Figure 4—figure supplement 2, providing a more comprehensive view.

4. Comment R3.4 (continued): The fitting algorithm is a little convoluted and difficult to understand now. I strongly suggest sharing the code of the model fitting and noise propagation online on a public repository.

We fully agree with the reviewer. To this end, we have updated the compiled code on the Digital.CSIC repository to its latest version. Additionally, we have uploaded the original.m scripts to the same repository, ensuring that our methodologies are open.

5. Comment E.2: The relation of true signal A and Rician-biased signal E(M) was explained in Koay and Basser JMR 2006, where Equation 13 shows {E(M)}^2^ ~ = A^2^ + σ_Rician^2^ at high SNR. The σ_Rician^2^ does not have a factor of 2. In the response of comment E.2, authors used the equation for E(M^2^). However, the magnitude signal is E(M), not sqrt{E(M^2^)}. Therefore, authors should cite and use the relations in Koay and Basser JMR 2006 to correct and estimate the SNR, though it may not affect the numerical results significantly.

We thank the reviewer for their comment. Indeed, for the previous version of the response to reviewers, we tried both approaches for the SNR calculation. Results were similar, but the method based on the mean shows some errors due to the iterative process, as illustrated in Author response image 6. Based on this result, we thus decided to use the method based on Aja-Fernandez et al.

**Author response image 6. sa2fig6:** Signal to noise ratio estimated for two slices using the proposed method on the left,and Koay’s on the right.

Elaborating a bit more on this topic, we acknowledge the issue raised regarding the relationship between the true signal A and Rician-biased signal E(M) as explained in Koay and Basser JMR 2006. This part of our paper was advised by Dr. Aja-Fernandez, who includes the referenced paper in his work. Koay's paper does use a mixed expression, and while {E(M)}^2^ converges to A^2^ + σ_Rician^2^ at high SNR, our approach is valid for all SNR levels.

Also, the reviewer is right about the bias of the signal. One must clearly specify what bias one wants to Signal to noise ratio estimated for two removes. In general (not always true), if we estimate A, slices using the proposed method on the we must remove the bias to E{M} and if we estimate left, and Koay’s on the right. A^2^, we must remove the bias to E{M^2^}. However, please note that in our case, our goal is not removing the bias, but rather defining an estimator for A. This estimator could have very different properties (bias, minimum error, edge protection, quick convergence, etc.). Nevertheless, in our case, since we are looking for a global estimation of the SNR, we use a standard simple approach that assumes that:

A2^= E{M2}−2σ2 This is a classic assumption known as the conventional approach (McGibney Med. Phys. 1985) and used by many authors in MRI. Therefore, we can define the estimator for SNR as SNR2^=A2^σ2=E{M2}−2σ2σ2 It is important to stress that this is a simple estimator, and not a method to *unbias* a result. Note that the purpose is not to obtain a detailed estimation of the original signal A (i.e., a perfect filtered and denoised version of the image), but an estimation reflecting the global levels of SNR. Alternatively, we could have followed the method proposed by the reviewer, that basically consists of using the mean of the Rician instead of the second-order moment:E{M}=π2L12 (−SNR22)σ

However, in this case, to use E{M}, we must approximate the function L_1/2_ for high SNR and use a recursive method. Experience tells us that, as the reviewer suggests, results would be very, very similar to the proposed approach for this rough estimation.

Finally, a more accurate estimation of the SNR, could also be obtained through a more accurate estimator of *A*, such as NLM filter, an EM approach such as the one in DeVore SPIE 2000, or any other statistical method. However, we fell that the method we chose is well accepted by the community, since it has been used in many different papers, for instance: Algarín*, Sci Rep* (2020); Afzali, MRM 2020; Manjon, Medical Image Analysis 2008.

Materials and methods:

“We also tested an alternative method (Koay and Basser, 2006) for SNR calculation. While the average SNR quantification was very similar between the two approaches, in the Koay and Basser SNR maps we observed artefacts due to the iterative process, so we decided to use the method by Aja-Fernandez et al. 2015.”